# VLForgery Face Triad: Detection, Localization and Attribution via Multimodal Large Language Models

**Xinan He**[1], **Yue Zhou**[2][*] **Bing Fan**[3], **Bin Li**[2], **Guopu Zhu**[4], **Feng Ding**[1][†]

[1]Nanchang University `shahur@email.ncu.edu.cn; fengding@ncu.edu.cn`
[2]Shenzhen University `2450042008@email.szu.edu.cn; libin@szu.edu.cn`
[3]University of North Texas `bingfan@my.unt.edu`
[4]Harbin Institute of Technology `guopu.zhu@hit.edu.cn`

## Abstract

Faces synthesized by diffusion models (DMs) with high-quality and controllable attributes pose a significant challenge for Deepfake detection. Most state-of-the-art detectors only yield a binary decision, incapable of forgery localization, attribution of forgery methods, and providing analysis on the cause of forgeries. In this work, we integrate Multimodal Large Language Models (MLLMs) within DM-based face forensics, and propose a fine-grained analysis triad framework called **VLForgery**, that can 1) predict falsified facial images; 2) locate the falsified face regions subjected to partial synthesis; and 3) attribute the synthesis with specific generators. To achieve the above goals, we introduce **VLF** (Visual Language Forensics), a novel and diverse synthesis face dataset designed to facilitate rich interactions between 'Visual' and 'Language' modalities in MLLMs. Additionally, we propose an extrinsic knowledge-guided description method, termed **EkCot**, which leverages knowledge from the image generation pipeline to enable MLLMs to quickly capture image content. Furthermore, we introduce a low-level vision comparison pipeline designed to identify differential features between real and fake that MLLMs can inherently understand. These features are then incorporated into EkCot, enhancing its ability to analyze forgeries in a structured manner, following the sequence of detection, localization, and attribution. Extensive experiments demonstrate that VLForgery outperforms other state-of-the-art forensic approaches in detection accuracy, with additional potential for falsified region localization and attribution analysis.

## 1 Introduction

AI-generated faces have enriched human life by creating realistic virtual characters that enhance our experiences in various creative industries. However, the domain of AI-generated face forensics is encountering unprecedented challenges due to the rapid advancement of generative models [8, 55, 49, 12, 24, 57]. In particular, diffusion models [38, 35] generate synthetic faces with substantially higher fidelity than earlier technologies such as variational autoencoders (VAEs) [41], rendering it increasingly challenging for detectors to identify subtle forgery artifacts. This development has led to a significant surge in the proliferation of forgeries, which present grave threats to personal privacy and security.

Several pioneering studies [3, 17, 44, 4, 48, 54] have investigated diffusion model (DM)-generated image forensics and demonstrated the effectiveness of their proposed models. However, a notable

---

[*]The author contributed equally to this work.

[†]Corresponding author

39th Conference on Neural Information Processing Systems (NeurIPS 2025).

limitation of these approaches is their lack of clear analysis for the underlying causes of the forgeries. Specifically, *most existing studies focus solely on detection, neglecting the localization of forgeries or attributing specific generation methods to them.*

Recently, multimodal large language models (MLLMs) have emerged as powerful tools for complex scene characterization.By leveraging extensive pre-training on both images and text, MLLMs can interpret visual tasks through natural language, providing a more nuanced understanding of forensic analysis [53, 14]. However, MLLM-based models face *two* primary limitations. **First**, forensic analysis research with MLLMs has largely focused on traditional face-swapping Deepfakes [36, 22], with limited analysis of DM-based synthetic faces. While recent work [53, 14] has shown that automated descriptions generated by MLLMs (*e.g.*, ChatGPT-4) or manual annotations guiding MLLM training, effectively capture artifacts in traditional face-swapping Deepfakes, these approaches still face challenges in generating credible forgery descriptions, particularly for high-fidelity DM-based images where subtle artifacts often evade detection. The primary issue lies in their reliance on subjective human-defined judgment biases to guide the generation of descriptions. Such descriptions may not align with the forgery knowledge that the models inherently understand. Additionally, MLLMs are typically trained for semantic-level visual alignment, lacking fine-grained forensic perception capabilities, which can lead to hallucinations in the generated descriptions. **Second**, none of the existing MLLM-based methods attempt fine-grained DeepFake forensics that simultaneously addresses *Detection*, *Localization*, and *Attribution*.

To address these challenges, we propose a triad framework-encompassing detection, localization, and attribution-designed for AI-generated face forensics, referred to as **VLForgery**. In the proposed framework, to address the lack of data specifically tailored for DM-based partial synthesis face, we create diverse prompt repositories and templates to build a new multimodal Visual Language Forensic (**VLF**) dataset, suitable for the triad tasks of detection, localization, and attribution.

Second, to enhance the reliability of MLLMs-generated descriptions, we introduce a low-level vision comparison pipeline that identifies low-level vision discrepancies between real and fake sample sets, guiding MLLMs to generate high-confidence descriptions. Building on this, we develop a description generation module, where we design an Extrinsic knowledge-guided Chain-of-thought (**EkCot**) method. This method also integrates additional knowledge from the generation pipeline, enabling the model to quickly grasp image content. Third, we develop a unified MLLMs fine-tuning and inference module, where we aggregate the generated images and corresponding descriptions to fine-tune MLLMs.

Specifically, VLForgery incorporates external information relevant to DM-based image generation with the low-level vision discrepancies between real and fake to structure a systematic analytical approach, allowing MLLMs to evaluate the image through the following steps: 1) *Detection*: identify whether the image is real or AI-generated; 2) *Localization*: For synthetic images, determine if the forgery is partial (only specific regions are altered) or full (the entire image is synthetic). For partial forgeries, locate the altered regions; 3) *Attribution*: Determine the likely method or model used to generate the forgery.

Our contribution can be summarized as follows:

- We investigate the potential of MLLMs in tackling AI-generated face forensics challenges by proposing the framework VLForgery. Additionally, we designed an extrinsic knowledge-guided chain-of-thought method, termed EkCot, that can assist the MLLMs in achieving fine-grained forensics performance.
- We introduce the VLF dataset, a new multimodal Visual Language Forensic dataset generated by diffusion models, designed to address the needs of fine-grained forensics tasks.
- We present a comprehensive evaluation experiment tailored to assess VLForgery on fine-grained forensic tasks. This evaluation encompasses three task scenarios, and 9 DM-based face types.

## 2 Related Work

### 2.1 DM-Generated Images

Recent advancements have centered on diffusion models [38], which have demonstrated exceptional performance in generating high-fidelity face images with high quality. Many studies have also

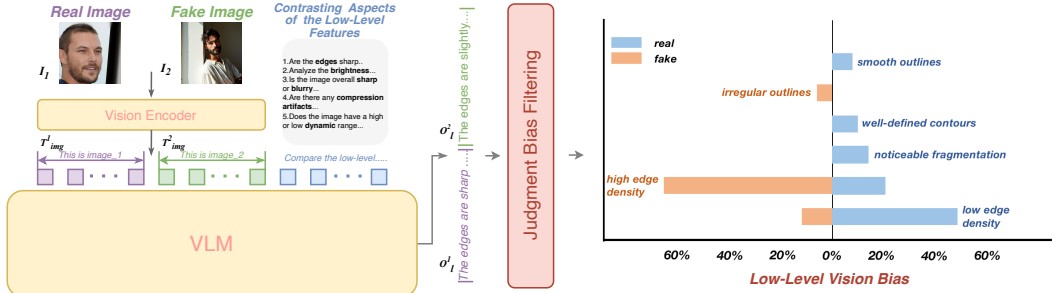

Figure 1: The comparison pipeline of low-level vision. By extracting the most distinctive low-level vision descriptors that differentiate the real and fake sample sets, we establish judgment biases for the MLLMs.

incorporated attention mechanisms to enhance the controllability of generated content [18, 23] such as text-guided [35, 30, 45, 32, 50, 2] and subject-guided [37, 21, 52, 16]. These achievements have led to the proposal of many DM-based datasets, such as object-oriented image datasets [56, 7, 39] and face-oriented image datasets [5, 24]. However, most DM-based partially synthesized face datasets suffer from poor quality and lack sufficient ground truth to support more fine-grained forensics tasks, such as forgery localization and attribution.

## 2.2   Fine-grained Forensics of DM-based Images

For Detecting tasks, traditional works usually attempt to find the traces left by diffusion models to expose synthesized images. For example, Chen *et al.* [3] proposed DRCT, a universal framework to enhance the generalizability of the existing detectors for detecting DM-based images. Recognizing the robust semantic comprehension capabilities of pre-trained vision-language models (*e.g.*, CLIP [34]), recent studies [17, 31] have demonstrated their promising results in detecting DM-generated images. For Locating tasks, a primary goal is for the detector to provide predictions with a localization map indicating which regions have been manipulated or to identify forgery artifacts within DM-based images. Guo et al. [10] proposed a hierarchical fine-grained Image Forgery Detection and Localization (IFDL) framework consisting of three components: a multi-branch feature extractor, localization, and classification modules. Zhang *et al.* [51] centers on detecting and segmenting artifact areas that are only perceptible to human observers, rather than identifying the full manipulation region. For Attributing tasks, the objective is to recognize the specific diffusion model that generates the images. Guarnera et al. [9] concentrate on attributing DM-generated images through a multi-level hierarchical approach. However, the aforementioned studies address the tasks of detection, localization, and attribute individually, and there is a notable absence of research that integrates these tasks into a unified framework.

## 2.3   Multimodal Large Language Models in Forensics

Recently, some studies have commenced utilizing MLLMs to investigate analytical ability within the field of forensics. Jia *et al.* [15] pioneered the exploration of the forensic capabilities of prompt engineering using ChatGPT. Zhang *et al.* [53] proposed incorporating common-sense reasoning to enhance traditional deepfake detection by manually labeling the manipulated regions and extending it to develop a Visual Question Answering (VQA) framework. Huang *et al.* [14] recently proposed an automatic method utilizing GPT-4o for annotations, thereby replacing human annotators. These studies primarily focus on traditional Deepfakes and are limited to single-forensic scenarios, lacking the capability to locate or attribute forgery areas in synthesized images. In the case of forgery artifacts from high-fidelity faces generated by diffusion models, generating credible forgery analyses may pose greater challenges. Ours, compared to other MLLM-based forensic frameworks, can guide MLLMs to generate highly credible fine-grained forgery descriptions and is applicable to a broader range of forgery types and more complex forensic scenarios.

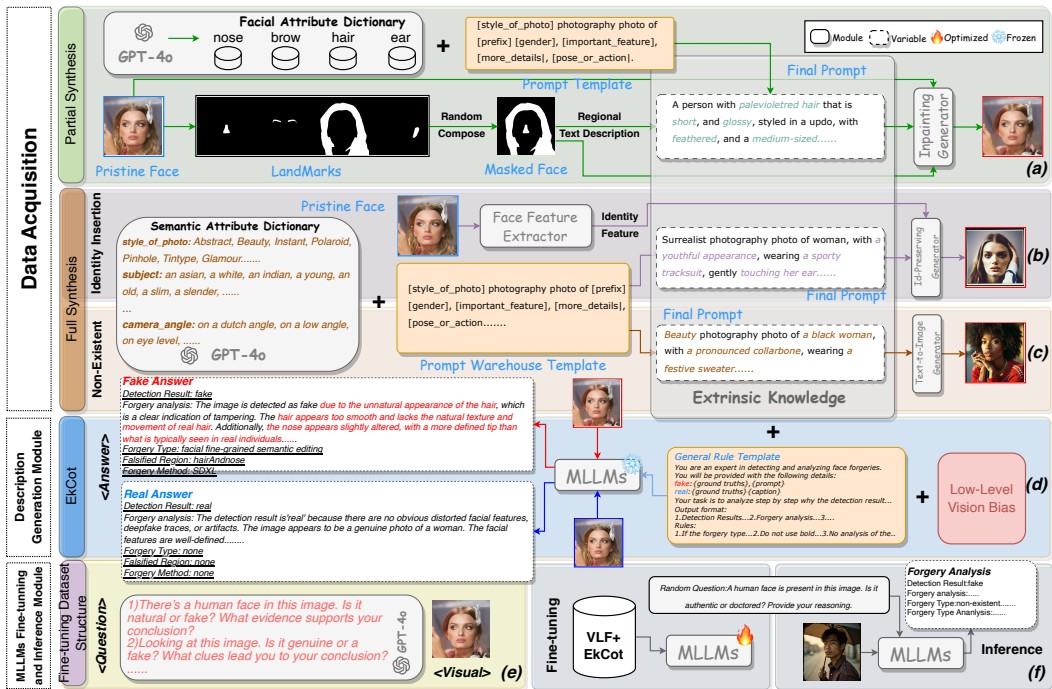

Figure 2: Construction of three modules for VLForgery framework: Data Acquisition, Description Generation Module, and MLLMs Fine-tuning and Inference Module.

## 3 VLForgery Framework

With the development of generative technologies, methods like diffusion models have emerged. Compared with traditional generative AI techniques (eg, VAE), the diffusion models enable partial synthesis guided by masks and text descriptions, as well as full synthesis through identity insertion or text descriptions. Therefore, our VLF dataset encompasses both partial and full synthesis types. In the following sections, we will demonstrate how to acquire their corresponding prompts and/or images. Furthermore, we will elaborate on the specific details of our proposed Description Generation Module and MLLMs Fine-tunning and Inference Module in VLForgery.

### 3.1 Data Acquisition

This section primarily outlines the pipeline details for collecting data. We synthesized all DM-generated data according to the requirements of the fine-grained forensics tasks. The details of the generation, involving partial and full synthesis, will be described in the following sections. *For a more detailed introduction to the dataset, please refer to Sec. B.*

#### 3.1.1 Partial Synthesis

In this part, we define the pipeline for partial synthesis. An exemplar pipeline is illustrated in Fig. 2(a).

To ensure the high quality of the partially synthesized images, we utilized the well-established facial dataset CelebAMask-HQ [19] as a data source since it already contains landmarks for various face attributes such as nose, brow, hair, ears, eyes, and teeth. Then, we use GPT-4o to generate each facial attribute dictionary and prompt template. For each image, we randomly combine masks to generate masked faces. Based on the generated masked face, we choose and combine templates for the corresponding attributes, then randomly select descriptive keywords from the relevant attribute dictionary to populate the templates, thereby generating the final prompt for image partial synthesis. During the image generation phase, we input the generated masked face, corresponding final prompt, and pristine face into the Inpainting Generator to produce an inpainted face.

### 3.1.2 Full Synthesis

We introduce two types of fully synthesized forgery: generating identity insertion (Fig. 2(b)) and non-existent faces (Fig. 2(c)). Both generation approaches utilize the same prompt template with slight modifications. In the face identity insertion pipeline, the objective is to preserve the original facial identity while situating it within a new scene. This pipeline initially uses a Face Feature Extractor to capture identity-specific features. In the generation of non-existent faces, the diffusion model creates images of completely new faces that are unrelated to any existing individual based on a given prompt.

## 3.2 Description Generation Module

This module leverages the extrinsic information required throughout the low-level vision comparison pipeline and the data acquisition processes to guide the construction of the forgery analysis description for each type of forgery sample in the VLF dataset. In this section, we introduce the workflows of the low-level vision comparison pipeline and the construction of EKCot separately. *We provide a detailed low-level vision analysis along with all the details and data you may need (refer to Sec C).*

### 3.2.1 Low-level Feature Comparison Pipeline

Motivation: In previous studies[46, 14], researchers attempted to generate explainable descriptions for Deepfakes using Vision Large Language Models. However, hallucinations are commonly observed in VQA tasks. Forensic analysis places more emphasis on the perception of image fine-grained details compared with general VQA tasks. Therefore, in the absence of ground-truth labels to supervise these explainable descriptions, **how can we determine their correctness?**

To address this challenge, we aim to seek forensic features that the model itself can understand to discriminate the authenticity of an image, thereby mitigating the model's struggles with explainable description generation. Therefore, we propose a low-level vision comparison pipeline that leverages the visual feature understanding capabilities of vision-large-language models (VLLMs) to identify low-level vision discrepancies between real and fake sample sets, as illustrated in Fig. 1. First, we randomly select one image from the real sample set and another from the fake sample set, as $I_1$ and $I_2$. Notably, the VLLMs is unaware of which image is real and which is fake. Furthermore, to comprehensively verify the low-level visual discrepancies between the real and fake sample sets, we selected ten distinct aspects and designed corresponding descriptive templates for each, as $T_{tem}$ (refer to Sec. C.1).

Initially, $I_1$ and $I_2$ are processed by the vision encoder $E_v$, which generates the corresponding token representations $T_{img}$. These tokens, along with the comparison template tokens $T_{tem}$, are then fed into the $VLLM()$ for generating low-level vision discrepancies text $O_l$. It is formulated as:

$$T_{img}^1, T_{img}^2 = E_v(I_1, I_2), \tag{1a}$$

$$O_l^1, O_l^2 = VLLM(T_{img}^1, T_{img}^2, T_{tem}). \tag{1b}$$

Judgment Bias filtering: Given $O_l^1, O_l^2$, we first perform tokenization and part-of-speech tagging to extract compound noun phrases. Next, we compute the proportion of each extracted compound noun phrase across all low-level vision discrepancy texts. Finally, we identify the top ten phrases with the largest proportion difference between the real and fake sample sets, selecting them as the judgment biases for the MLLMs. These phrases are then used to generate the final usable judgment bias descriptors through a predefined template (refer to Sec. C.3).

Furthermore, to avoid potential model interpretation biases, we selected the Llama-3.2-11B-Vision[1] model as the VLLM for the low-level vision comparison pipeline, ensuring consistency with the models used in subsequent stages.

### 3.2.2 Construction of EkCot

As shown in Fig. 2(d), we utilized MLLMs as the description generation model to construct a chain of thought for forgery analysis, termed EkCot. Initially, we propose a General Rule template designed to integrate the extrinsic information required for generating desired data. The extrinsic information includes the prompts used within the pipeline, the corresponding ground truth for the generated image, and the low-level vision judgment bias, which are subsequently combined to serve as prompt inputs for MLLMs. The ground truth encompasses several categories, including real/fake, forgery type,

forgery region, and forgery method. For pristine images, the forgery type, forgery region and forgery method values are set to 'none' by default. Additionally, image original features are provided as visual inputs to the MLLMs. Furthermore, we utilize the Llama-3.2-11B-Vision[1] as the description generation model.

## 3.3 MLLMs Fine-tuning and Inference Module

In this module, we designed the final fine-tuning data structure and the details of MLLMs fine-tuning. **Final Fine-tuning Data Structure.** The final fine-tuning data is composed in the form of triplets:

<visual, question, answer>. For question's format, we use ChatGPT-4o to generate a range of

question formats, as shown in Fig. 2(e), for example: 'Is this image real or fake? Can you provide the reasoning behind your judgment?' For each triplet, the question is presented in a randomly generated format. Additionally, each image is linked to its corresponding specific description (i.e., the answer). For the answer's format, we designed it to address three primary forensic tasks: detection,

localization, and attribution. The answer format follows this pattern: *'Detection Result: real/fake'* + $A_{result}$ + *'Forgery Type'* + *'Falsified Region'* + *'Forgery Method'*, where $A_{result}$ denotes the analysis of detection results. For analysis steps, our objective is to guide the model to adopt a chain of thought,

performing each step sequentially as follows: 1)Real or Fake Judgment: The authenticity of the image is first assessed. If an image is identified as a forgery, the type of forgery is then classified into one of three categories: partial synthesis, identity insertion, and non-existent faces. The reasons for the detection result are further analyzed. 2) Localization of Falsified Region: If the forgery involves partial synthesis, the specific regions of manipulation (*e.g.*, nose, brow, hair, ear) are identified. In cases involving multiple edits (*e.g.*, hairAndnose), each forged region is localized. 3) Attribution of Synthesis Face Generator: The final step determines the generator of the synthesis face. **Fine-tunning**

**and Inference Process.** We fine-tune the MLLMs on the final fine-tuning data through a structured two-step process (Fig. 2(f)). Inspired by LLaVA [25], we adopt their published Llava-1.5-7B model and training architecture. First, we fine-tune a projector to align the facial visual features extracted from the frozen CLIP visual encoder with the corresponding question text features. This alignment links the synthetic face's forgery artifacts with corresponding ground truth and detailed forgery descriptions. Second, we employ Low-Rank Adaptation (LoRA) [13] (rank=128, alpha=256) to selectively update the model by adjusting only the LoRA parameters, effectively fine-tuning the pre-trained language model.

During inference, the input is structured as a tuple <image, question>. For the questions, we continue to use ChatGPT-4 to generate diverse question formats, ensuring input variety. Finally, the model's response is structured, similar to the final fine-tuning data's answer format.

This approach ensures the model conducts forensic analysis on input faces in a structured, stepwise manner according to a pre-defined reasoning framework, thereby improving both its fine-grained forgery analysis and forensic accuracy.

Table 1: VLF dataset.

| Type | Source | Samples | |
|---|---|---|---|
| | | train | test |
| Pristine | FFHQ | 42k | 14k |
| | FF++ (original) | 12k | 4.5k |
| | CelebA-HQ | 18k | 6k |
| DM-based Partial Synthesis | SDXL | 76k | 19k |
| | SD2 | 76k | 19k |
| | Kandinsky2.2 | 76k | 19k |
| DM-based Full Synthesis | SDXL | 24k | 6k |
| | SD2 | 24k | 6k |
| | SD3 | 24k | 6k |
| | Kandinsky2.2 | 24k | 6k |
| | Flux | 24k | 6k |
| | InstantID | 8.1k | 2.5k |

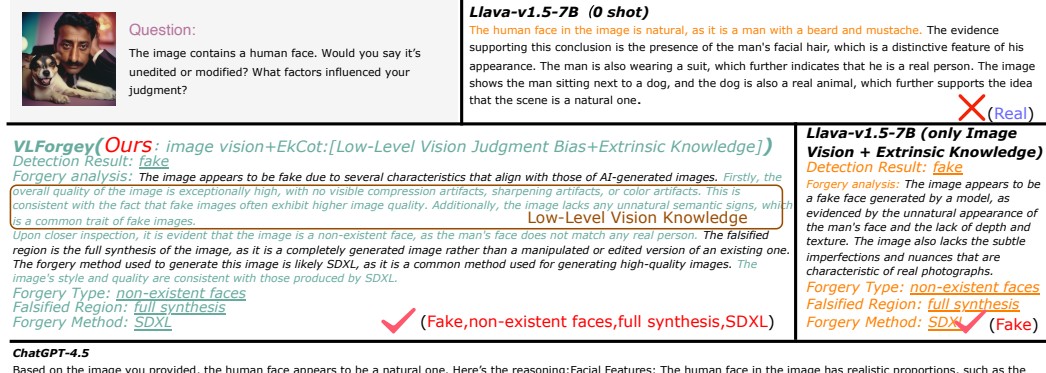

Figure 3: Qualitative results. The upper left presents a synthesis face from full synthesis. Ours is presented in contrast to the responses of Llava-v1.5-7B (0 shot), Llava-v1.5B-7B (conditional fine-tune), and Chatgpt-4.5, respectively. The lower-right bracket provides the result annotations.

# 4 Experiment

## 4.1 Experimental Settings

**Datasets.** All facial samples in experiments were sourced from the VLF. The VLF dataset consists of 12 subsets, with each subset divided into training and test sets in a 4:1 ratio, as illustrated in Table 1.

**Compared Baselines.** We selected four types of models for evaluation: 1) Naive convolutional neural networks (CNNs): Xception [6], Resnet-50 [11]. 2) Typical frequency detectors: SPSL [26], F3Net[33], SRM[29], NPR[40]. 3) Typical Spatial detectors: GramNet[28], SAFE[20], CNNspot[43], DRCT [3]. 4) VLLM-based method:CLIPping [17], Llava-1.5-7B[25], Qwen2.5-VL-7B[47], Llama3.2-11B-Vision[1].

**Implementation Details.** All experiments are based on the PyTorch and trained with 8 NVIDIA RTX L40. For training, we utilized the Adam optimizer with a learning rate of 2e-5 and a batch size of 128 for 3 epochs.

Table 2: Evaluation on Full Synthesis type. Results on different training and testing subsets using VLForgery. Accuracy is used for evaluation.

| Train | Full Synthesis(%)↑ | | | | | | Avg(%)↑ |
|---|---|---|---|---|---|---|---|
| | Non-existent | | | | | Id-insertion | |
| | SDXL | SD2 | SD3 | Kandinsky2.2 | Flux | InstantID | |
| SDXL | 99.35 | 98.47 | 96.02 | 98.78 | 98.7 | 87.38 | 96.45 |
| SD2 | 99.47 | **99.97** | 98.38 | 99.40 | 98.92 | 80.67 | 96.13 |
| SD3 | 99.65 | 97.23 | **99.98** | 97.95 | 99.70 | 21.93 | 86.07 |
| Kandinsky2.2 | **99.95** | 99.37 | 97.95 | **99.95** | 98.83 | 93.75 | **98.44** |
| Flux | 98.01 | 95.85 | 98.10 | 98.25 | **100.00** | 48.10 | 74.89 |
| InstantID | 90.38 | 87.05 | 60.83 | 87.40 | 84.97 | 92.97 | 83.94 |

Table 3: Evaluation on Partial Synthesis type. Results on different training and testing subsets using VLForgery. Accuracy is used for evaluation.

| Train | Partial Synthesis(%)↑ | | | Avg(%)↑ |
|---|---|---|---|---|
| | SDXL | SD2 | Kandinsky2.2 | |
| SDXL | **80.35** | **68.78** | 98.08 | **82.40** |
| SD2 | 54.53 | 61.77 | 93.01 | 69.77 |
| Kandinsky2.2 | 42.09 | 34.20 | 95.93 | 57.41 |

## 4.2 Evaluation of Multi-task Forensics

### 4.2.1 Task1: Detecting the Authenticity of Faces

Table 4: Cross-generator dataset evaluation on ACC metric. All methods are trained on SDXL and evaluated on other subsets. ∗ indicates the use of trained models provided by the authors. Note that 'PS' means Partial Synthesis, and 'FS' means Full Synthesis.

| Type | Method | Train–SDXL(%)↑ | | SD2(%)↑ | | Kandinsky2.2(%)↑ | | SD3(%)↑ | Flux(%)↑ | InstantID(%)↑ | Avg(%)↑ |
|---|---|---|---|---|---|---|---|---|---|---|---|
| | | PS | FS | PS | FS | PS | FS | | | | |
| Naive | Resnet-50[11] | 35.09 | 99.73 | 11.87 | 95.67 | 58.40 | 97.37 | 90.58 | 94.03 | 94.58 | 75.26 |
| | Xception[6] | 68.19 | 99.98 | 40.10 | 97.58 | 96.89 | 99.68 | 97.72 | 99.03 | 99.57 | 88.75 |
| Frequency | SPSL[26] | 31.62 | 99.98 | 4.79 | 96.92 | 51.42 | 99.83 | 96.92 | 99.28 | 95.09 | 75.09 |
| | F3Net[33] | 74.09 | 99.91 | 46.97 | 99.45 | 81.65 | 99.90 | 99.33 | 99.98 | 99.84 | 89.01 |
| | SRM[29] | 59.21 | 99.95 | 12.30 | 98.65 | 98.34 | 99.57 | 98.43 | 98.17 | 97.06 | 84.63 |
| | NPR[40] | 42.67 | 99.98 | 10.22 | 98.98 | 70.51 | 99.57 | 98.25 | 99.75 | 99.37 | 79.92 |
| Spatial | GramNet[28] | 34.17 | 99.98 | 14.28 | 96.93 | 48.37 | 99.65 | 93.43 | 99.27 | 88.45 | 74.94 |
| | SAFE[20] | 56.47 | 82.08 | 54.08 | 81.55 | 56.32 | 78.98 | 88.43 | 57.05 | 71.43 | 61.66 |
| | CNNspot[43] | 26.30 | 97.93 | 9.04 | 73.00 | 26.97 | 74.12 | 39.95 | 70.00 | 73.60 | 54.55 |
| | DRCT*[3] | 57.65 | 99.41 | 58.52 | 97.71 | 99.33 | 99.92 | 65.97 | 59.06 | 98.92 | 81.83 |
| VLLM-Based | CLIPping *Adapter*[17] | 36.29 | 99.91 | 32.13 | 94.37 | 75.04 | 98.38 | 98.58 | 97.95 | 97.05 | 81.03 |
| | CLIPping *Linear Probing*[17] | 61.79 | 99.98 | 54.33 | 97.18 | 87.33 | 99.65 | 97.88 | 99.03 | 99.80 | 88.55 |
| | Llava-1.5-7B[25] | 69.92 | 99.92 | 51.07 | 99.97 | 99.56 | 99.74 | 97.66 | 99.12 | 99.69 | 87.07 |
| | Qwen2.5VL-7B*[47] | 4.77 | 4.32 | 3.78 | 21.88 | 31.92 | 6.55 | 3.82 | 3.00 | 46.13 | 14.02 |
| | Llama-3.2-11B-Vision*[1] | 53.46 | 64.97 | 52.13 | 68.95 | 58.42 | 69.63 | 47.80 | 63.05 | 58.49 | 59.66 |
| | VLForgery (ours) | 78.62 | 99.98 | 66.32 | 99.97 | 99.96 | 99.97 | 99.98 | 99.97 | 99.57 | **93.82** |

Table 5: Evaluation of falsified regions localization performance across distinct generators. Accuracy is used for evaluation.

| Train | hair(%)↑ | | | Avg(%)↑ | brows(%)↑ | | | Avg(%)↑ | ears(%)↑ | | | Avg(%)↑ | nose(%)↑ | | | Avg(%)↑ |
|---|---|---|---|---|---|---|---|---|---|---|---|---|---|---|---|---|
| | SDXL | SD2 | Kandinsky2.2 | | SDXL | SD2 | Kandinsky2.2 | | SDXL | SD2 | Kandinsky2.2 | | SDXL | SD2 | Kandinsky2.2 | |
| SDXL | **88.99** | 86.91 | **92.44** | **91.13** | **73.84** | 53.36 | 72.61 | **66.60** | **59.16** | 51.99 | 72.01 | 61.05 | 77.25 | 56.53 | 91.34 | 75.04 |
| SD2 | 79.68 | 82.63 | 85.42 | 82.58 | 65.38 | **62.64** | 66.84 | 64.95 | 40.30 | 50.33 | 52.66 | 47.76 | 58.46 | **75.45** | 86.67 | 73.53 |
| Kandinsky2.2 | 58.53 | 50.47 | 85.79 | 64.93 | 34.33 | 13.67 | **79.38** | 42.46 | 29.76 | 24.98 | 71.61 | 42.12 | 14.01 | 16.11 | 89.04 | 39.72 |

*Intra-type performance with MLLMs.* In this task, we first evaluate the Full Synthesis type performance with VLForgery. As shown in Table 2, the Full Synthesis type comprises two subtypes: Non-existent and Id-insertion. The non-existent face subtype consists of five distinct subsets, each corresponding to a specific generator. For the non-existent subtype, training and testing within each subset surpass 99.35% accuracy, while cross-subset testing achieves an average accuracy of 91.66%. Conversely, cross-subtype performance varies significantly. For instance, training on the Flux subset and testing on InstantID results in a classification accuracy drop to 48.1%.

Table 3 presents the performance evaluation of the Partial Synthesis type. This type consists of three subsets, each associated with a specific generator. Training was conducted on each of the three subsets, and testing was subsequently performed across all subsets. As indicated in Table 3, training on partial synthesis samples generated by SDXL-based models yields superior generalization performance.

Based on these observations, detectors trained with full-synthesis images are typically trained based on global features, focusing on the overall structure and content of the image. Also, they primarily focus on analyzing the integrity of the entire image. However, partial synthesis images often involve modifications to small areas of the image, and these details may not be obvious in the global features. Therefore, when these detectors encounter partial synthesis images with only a small local area tampered with, they may fail to capture the subtle changes, leading to a reduced recognition rate for local tampering.

*Cross-Generator Faces Classification.* We propose evaluating the generalization performance of distinct generators, encompassing both Full Synthesis and Partial Synthesis types. As illustrated in Table 4, all detectors were trained on samples with partial synthesis and full synthesis from the SDXL-based model, and subsequently tested on other subsets. For each compared method, we trained for 10 epochs and validated their performance on the final epoch. VLForgery exhibits a notable advantage, especially in accurately detecting partially synthesized face images

### 4.2.2 Task2: Locating the Falsified Regions

We evaluate the localization capabilities of VLForgery for four distinct regions (*i.e.*, nose, brow, hair, and ears), as shown in Table 5. The Partial Synthesis type contains edited faces with single-edit and

Table 6: Evaluation of model's attribution accuracy on VLF.

| Method | FS_SDXL | FS_SD2 | FS_SD3 | FS_IID | FS_Kan2.2 | FS_Flux | PS_SDXL | _PS_SD2 | PS_Kan2.2 | Avg. |
|---|---|---|---|---|---|---|---|---|---|---|
| resnet50 | 99.93 | 99.53 | 84.43 | 98.82 | 99.23 | 99.87 | 11.81 | 6.09 | 76.48 | 75.13 |
| xception | 99.98 | 99.97 | 87.93 | 99.57 | 99.37 | 99.52 | 16.87 | 27.64 | 91.81 | 80.30 |
| efficientb4 | 99.99 | 99.88 | 88.93 | 98.15 | 98.95 | 99.90 | 27.61 | 27.85 | 81.43 | 80.29 |
| Guarnera *et al.* | 99.78 | 99.87 | 86.65 | 97.72 | 99.92 | 99.43 | 35.75 | 12.71 | 90.17 | 80.22 |
| VLForgery | 93.27 | 92.47 | 93.24 | 90.42 | 89.13 | 82.43 | 63.64 | 60.20 | 73.45 | 82.03 |

Table 7: Ablation Study of the VLForgery's forgery description generation component in the detection task. 'VF', and 'EkCot' represent visual input features in forgery description generation, and extrinsic knowledge-guided chain-of-thought, respectively

| Method | | | Detection(%)↑ | | | | | | | | |
|---|---|---|---|---|---|---|---|---|---|---|---|
| | | | SDXL | | SD2 | | Kandinsky2.2 | | SD3 | Flus | InstantID |
| Name | VF | EKCot | Partial Synthesis | Full Synthesis | Partial Synthesis | Full Synthesis | Partial Synthesis | Full Synthesis | | | |
| VariantA | | | 69.92 | 99.92 | 51.07 | 99.97 | 99.56 | 99.74 | 97.66 | 99.12 | 99.69 |
| VariantB | | ✓ | 68 | 99.99 | 47.64 | 99.98 | 99.18 | 99.99 | 99.98 | 99.98 | 99.84 |
| Ours | ✓ | ✓ | 78.62 | 99.98 | 66.32 | 99.97 | 99.96 | 99.97 | 99.98 | 99.97 | 99.57 |

Table 8: Ablation Study of the VLForgery's forgery description generation component in the localization task. 'VF', and 'EkCot' represent visual input features in forgery description generation, and extrinsic knowledge-guided chain-of-thought, respectively.

| Method | | | Localization(%)↑ | | | | | | | | | | | | |
|---|---|---|---|---|---|---|---|---|---|---|---|---|---|---|---|
| | | | hair | | | brows | | | ears | | | nose | | |
| Name | VF | EKCot | SDXL | SD2 | Kandinsky2.2 | SDXL | SD2 | Kandinsky2.2 | SDXL | SD2↑ | Kandinsky2.2 | SDXL | SD2 | Kandinsky2.2 |
| VariantA | | | 54.41 | 55.06 | 59.49 | 38.63 | 33.10 | 46.76 | 22.56 | 18.44 | 30.46 | 38.63 | 28.10 | 42.82 |
| VariantB | | ✓ | 88.77 | 78.51 | 97.48 | 66.09 | 39.44 | 75.61 | 45.25 | 34.86 | 74.56 | 40.36 | 46.95 | 96.13 |
| Ours | ✓ | ✓ | 88.99 | 86.91 | 92.44 | 73.84 | 53.36 | 72.61 | 59.16 | 51.99 | 72.01 | 77.25 | 56.53 | 91.34 |

multi-edit. Additionally, this type is categorized into three subtypes based on distinct generators. Each subtype was trained separately, and tested across all subtypes, with validation results tallied across the four facial regions. Note that, if a multi-edited face is detected with the result 'earAndnose', but the ground truth is 'earAndhair', the detection of the ear region is marked as correct, while the hair region is marked as incorrect.

*Why Use Natural Language Instead of Masks for Localization Results?* Fig. 4 illustrated a comparison of localization results between VLForgery and PSCC-Net (IFDL method). For subtle falsified facial regions, such as brows and nose, IFDL methods often struggle to identify them accurately, being easily influenced by extraneous features from the image background, which leads to incorrect localization. In contrast, for facial localization, where tampered areas are limited, VLForgey leverages natural language descriptions instead of mask images to reduce the difficulty of localization.

This approach significantly enhances localization accuracy and minimizes susceptibility to interference from background features. Furthermore, to quantify the performance gap in localization between ours and the IFDL method, we selected 1,000 samples from the Partial Synthesis test set. We used the IFDL method to generate mask images, which were then manually evaluated. A localization was deemed correct if any part of the manipulated area was slightly visible in the mask image; otherwise, it was considered incorrect. As shown in Table 9, our method demonstrates a clear advantage in accurately localizing manipulated regions.

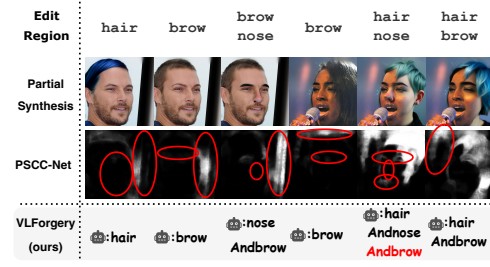

Figure 4: Comparative results of tamper facial localization capabilities between IFDL method and VLForgery. Both models were trained on partial synthesis generated by SDXL, and tested on Kandinsky2.2 samples. Mislocalized areas are marked in red.

*Cross-Generator Falsified Region Localization.* As observed in Table 5, localization accuracy for hair consistently remains the highest, followed by the nose, brows, and then the ears. For instance, when trained and tested on SDXL samples, the hair localization accuracy reached 88.99%. In contrast, localization accuracy for smaller regions, such as the ears, decreases to 59.16%.

Table 9: Evaluation of falsified regions localization performance across distinct generators with VLForgery and IFDL method.

| Method | SD2(%)↑ | | | | Kandinsky2.2(%)↑ | | | |
|---|---|---|---|---|---|---|---|---|
| | hair | brows | ears | nose | hair | brows | ears | nose |
| PSCC-Net [27] | 53.57 | 39.47 | 41.20 | 29.73 | 60.71 | 68.42 | 63.16 | 44.44 |
| VLForgery | **86.91** | **53.36** | **51.99** | **56.53** | **92.44** | **72.61** | **72.01** | **91.34** |

### 4.2.3 Task3:Attributing Source Models of Synthesis Faces

In this task, we focus on evaluating the VLForgery's capability to capture the synthesis semantic patterns of different generators. We fine-tuned VLForgery on the VLF training set, which includes all subsets, and tested its performance across each generator for the two synthesis types. As shown in Tab. 6, the attribution performance for full synthesis is superior, with an average accuracy of 91.71%. However, attribution accuracy for partial synthesis falls below 70%. We attribute this disparity to the detection difficulty of partial synthesis.

### 4.3 Ablation Study

Two main factors affect VLForgery forensics performance: (1) the impact of EkCot compared to outputting only forgery detection results, and (2) the influence of additional visual input features on forensics performance in EkCot. We conducted ablation studies for both detection and localization tasks. Table 7 presents the ablation results for the detection task, demonstrating a significant performance improvement in detecting partially synthesized images when using the EkCot. Table 8 displays the ablation results for the localization task, where the model's localization capability significantly decreased without EkCot. And compared to the forgery description generation method without visual assistance, the model performed better on SDXL and SD2 samples, though it showed some limitations on Kandinsky 2.2 generated samples.

### 4.4 Qualitative Study

The primary advantage of the multimodal large language models (MLLMs) is their flexible output, allowing them to provide a comprehensive analysis of the synthesized images. As shown in Fig. 3, we compared VLForgery with some existing MLLMs, demonstrating that VLForgery enables detailed capabilities, including forgery analysis, localization of falsified regions, and synthesis attribution.

## 5 Conclusion

In this paper, we introduce VLForgery, a framework designed for AI-generated faces in fine-grained forensics scenarios. To address the lack of partially synthesized face datasets, we constructed VLF. Additionally, we introduce EkCot, which provides a fine-grained analysis of forgery artifacts. The satisfying performance of the proposed framework is justified by extensive evaluations.

## Acknowledgments and Disclosure of Funding

This work was supported in part by the National Natural Science Foundation of China (Grant U23B2022, U22B2047, 62262041, 62172402), the Shenzhen R&D Program (Grant JCYJ20250604181211016, SYSPG20241211174032004), and the Jiangxi Provincial Natural Science Foundation (Grant 20232BAB202011).

We also sincerely thank Shu Hu, from the Department of Computer and Information Technology, Purdue University, IN, 46202, USA, for providing insightful advice and support during the preparation of this manuscript.

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

## Structure of Contents

## A    Additional Introduction

### A.1    Limitations

Most existing detectors for AI-generated content can achieve relatively good generalization on fully synthesized images with minimal training data. However, as discussed in the main text, we introduced additional samples in VLF partial synthesis. These samples incorporate real features from the pristine images to confuse detectors and evade detection. Even when these partially synthesized images are included in the training set, many binary classifiers still suffer from performance degradation. Although our VLF framework achieves the best generalization performance, it still falls short of our desired outcomes. Furthermore, the cross-generalization ability between different types of models, such as training on full synthesis and testing on traditional face-swapping deepfakes, is inherently limited due to differences in the generation processes. Enhancing generalization for more unknown generation methods remains a field worth exploring.

Moreover, for the VLF dataset, different generators and varying prompts may leave distinct forgery semantics. It is necessary to expand the dataset and evaluate the detection and generalization performance across these different forgery semantics.

### A.2    Future Works

First, we plan to expand the VLF dataset to encompass a broader range of forgery types, generators, and a more diverse and randomized set of semantic attributes. Second, we seek to enhance the model's generalization across varying forgery types by developing more robust model architectures and investigating more reliable techniques for generating forgery descriptions.

## B    Details of VLF Dataset Construction

### B.1    Details of Partial Synthesis Pipeline

In this part, we define the pipeline for partial synthesis, which includes three steps: 1) Mask Preparation; 2) Prompt Design; 3) Image Generation.

**Mask Preparation**. To ensure the high quality of the partially synthesized images, we utilized the well-established facial dataset CelebAMask-HQ [19] as a data source since it already contains landmarks for various face attributes such as nose, brow, hair, ears, eyes, and teeth. However, to improve the robustness of our later fine-grained analysis framework for more highly realistic generated images, we only focus on nose, brow, hair, and ears landmarks since they can usually be

generated more realistically. Other attributes such as eyes and teeth, are often generated with more obvious artifacts like misaligned pupils or incorrectly positioned teeth, even with specific prompt design. Therefore, we ignore them during the generation phase but keep their original appearance in the generated image.

**Prompt Design**. For the selected attributes, we use GPT-4o to generate each facial attribute dictionary and prompt template. For each image, we randomly combine masks to generate masked faces. Therefore, each masked face may contain $i \in \{1, 2, 3, 4\}$ masks. Based on the generated masked face, we choose and combine templates for the corresponding attributes, then randomly select descriptive keywords from the relevant attribute dictionary to populate the templates, thereby generating the final prompt for image partial synthesis.

**Image Generation**. During the image generation phase, we input the generated masked face, corresponding final prompt, and pristine face into the Inpainting Generator to produce an inpainted face. For the Inpainting Generator, several widely used open-source models were selected: SDXL, SD2, and Kandinsky2.2. Furthermore, to better simulate real-world scenarios, certain forgery methods extract modified regions and overlay them onto the corresponding area of the pristine image, preserving the unedited portions of the original. These retained portions can then be leveraged to interfere with forensic detectors, thereby facilitating evasion. Consequently, we included this category of partial synthesis in our dataset.

## B.2  Details of Full Synthesis Pipeline

**Prompt Design**. With advancements in generative models, particularly text-guided models that now support longer input prompts to deliver more diverse outputs. In this context, we propose a semantic attribute dictionary, created with guidance from ChatGPT-4o, specifically designed to support the generating of realistic faces across diverse real-world scenarios. And, we further devised a corresponding prompt template, (*i.e.*, Prompt Warehouse Template, as shown in Fig. 2(b, c)). By encompassing diverse real-world scenarios and stylistic elements, this template is structured to simulate realistic facial characteristics across varied photographic conditions.

**Image Generation**. We introduce two types of fully synthesized forgery: generating identity insertion (Fig. 2(b)) and non-existent faces (Fig. 2(c)). Both generation approaches utilize the same prompt template with slight modifications. In the face identity insertion pipeline, the objective is to preserve the original facial identity while situating it within a new scene. This pipeline initially uses a Face Feature Extractor to capture identity-specific features. These identity features are then integrated with the final prompt template and input into the ID-preserving Generator to produce the synthesized image. InstantID [42] was chosen as the model of choice, with all identities sourced from the CelebAMask-HQ [19] facial datasets. In the generation of non-existent faces, the diffusion model creates images of completely new faces that are unrelated to any existing individual based on a given prompt. In this case, the final prompt is created by randomly selecting keywords from the semantic attribute dictionary and inserting them into the prompt template, which then serves as the sole input to the Text-to-Image Generator. For the Text-to-Image Generator, we selected several open-source models that support extended text inputs, including Flux, SD3, Kandinsky2.2, SD2, and SDXL.

## B.3  Details of Attribute Dictionary

**Facial Attribute Dictionary:** For fine-grained facial manipulations, the dictionary mainly includes the following attributes:

- *[size]*: Defines the relative size of facial features.
- *[shape]*: Specifies the contour and structure of facial features.
- *[color]*: Describes the color characteristics of facial elements.
- *[styles]*: Indicates the stylistic choices applied to facial features.
- *[description]*: Provides a textual description of the face.
- *[additional_features]*: Lists extra details, such as freckles, scars, wrinkles, or facial expressions, that add realism to the modifications.

The data source of the facial attribute dictionary is shown in Fig. B.1 (a).

**Semantic Attribute Dictionary:** For semantic attribute dictionary, including following attributes:

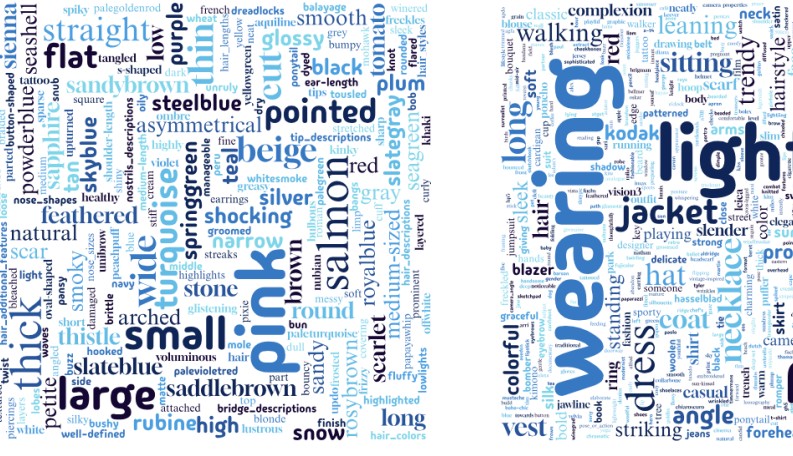

| (a) Facial Attribute Dictionary | (b) Semantic Attribute Dictionary |

Figure B.1: The data source of facial attribute dictionary and semantic attribute dictionary.

- *[style_of_photo]*: Defines the overall stylistic direction of the image.
- *[subject]*: Specifies the characteristic attributes of the subject.
- *[important_feature]*: Important attributes of the subject.
- *[more_details]*: Further attributes of the subject.
- *[pose_or_action]*: Interaction between the subject and the scene.
- *[framing]*: Indicates whether body features are included.
- *[lighting]*: Lighting conditions in the scene.
- *[camera_angle]*: Camera shooting angle.
- *[camera_properties]*: Different parameters produce varying focus effects.
- *[photographer]*: Similar to camera parameters, the photographer's choices affect the presentation of the image.

The data source of each attribute is shown in Fig. B.1 (b).

## B.4 Examples of VLF Dataset

We selected a subset of samples from the VLF dataset and organized them by categorizing based on different generators and detailed characteristics.

Fig. B.2 illustrates the partial synthesis generated by different generators guided by the facial attribute dictionary.

Fig. B.3 illustrates the full synthesis generated by different generators guided by the semantic attribute dictionary.

## C  Details of Low-Level Vision Comparison Pipeline

In this section, we elaborate on three aspects: low-level vision prompt design, low-level vision description analysis, and low-level vision bias analysis.

### C.1  Design of Low-Level Vision Prompt

Due to the high fidelity of existing DM-based images, it is challenging to explore subtle differences between them and real images from a purely visual perspective. However, images generated by different generators still exhibit certain common characteristics that distinguish them from real images. We refer to these characteristics as low-level vision biases.

One key challenge is determining the specific aspects in which these low-level vision biases manifest, such as lighting, noise, or color distribution. To address this, we investigate these biases from the following 10 low-level perspectives:

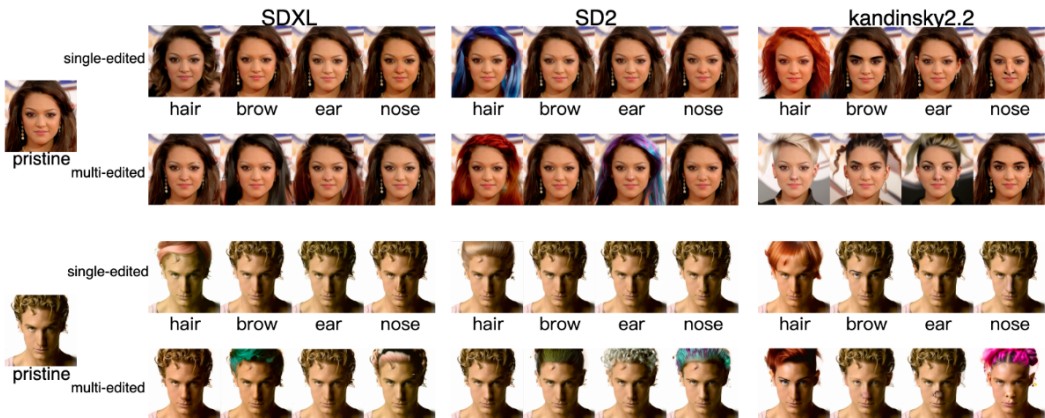

Figure B.2: Examples of Partial Synthesis in VLF.

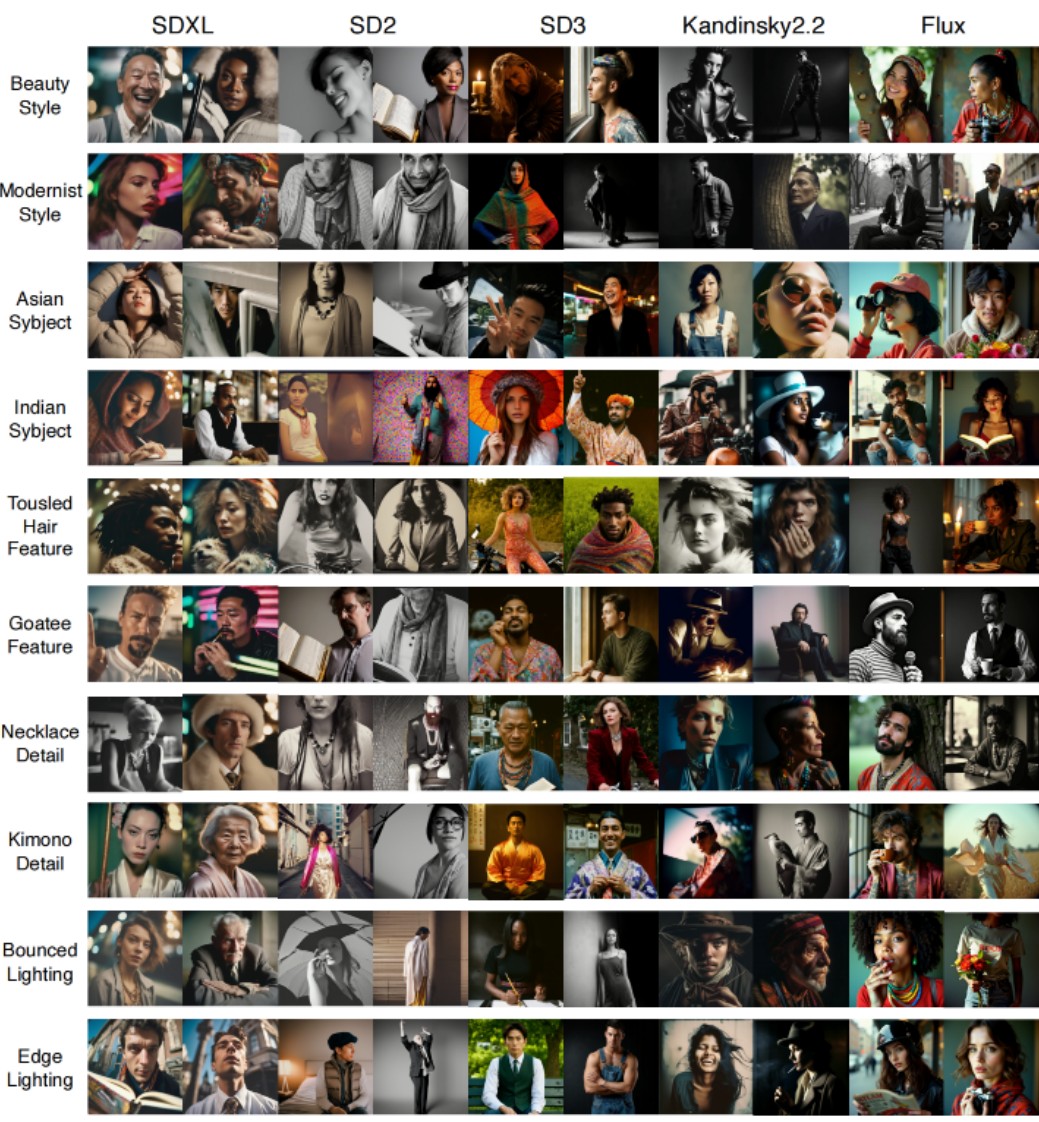

Figure B.3: Examples of Full Synthesis in VLF.

**10 Perspectives:**
- Are the edges sharp, blurry, or fragmented? Are the outlines complete or irregular? Are there any areas with particularly high or low edge density? Do the edges follow a specific direction (e.g., horizontal, vertical, diagonal)? Provide specific examples of edge and contour characteristics in different parts of the image.
- Analyze the brightness and contrast of the image in detail. Is the image overall too bright, too dark, or well-balanced? Are there any areas with extreme highlights or shadows? Describe the distribution of contrast across the image. Are there any signs of overexposure or underexposure? Provide specific examples of brightness and contrast variations in different parts of the image.
- Is the image overall sharp or blurry? If blurry, what type of blur is present (e.g., motion blur, Gaussian blur, defocus blur)? Are there specific areas that are more blurred than others? Compare the sharpness of different regions in the image. Provide specific examples of sharp and blurry areas.
- What are the dominant colors, and how are they distributed across the image? Are the colors evenly distributed, or are there any anomalies like oversaturation, color distortion, or color blocks? Analyze the brightness and saturation of colors in different parts of the image. Provide specific examples of color characteristics and anomalies.
- Are there any geometric distortions, such as stretching, warping, or asymmetry? Are there any compression artifacts, like blocky patterns or jagged edges? Analyze the symmetry of the image. Are there any unnatural deformations? Provide specific examples of distortions and anomalies in different parts of the image.
- Is the image noisy? If so, what type of noise is present (e.g., Gaussian noise, salt-and-pepper noise, quantization noise)? Describe the intensity and distribution of noise across the image. Analyze the texture complexity—is it uniform or varied? Are there any directional patterns in the texture? Provide specific examples of noise and texture characteristics in different parts of the image.
- Is the lighting uniform or uneven? Are the shadows natural and consistent with the lighting direction, or do they appear unnatural? Compare the highlights and shadows in different parts of the image. Are there any lighting anomalies, such as overly bright or dark areas? Provide specific examples of lighting and shadow characteristics.
- Are the details rich and well-defined, or are they lacking? How does the resolution affect the clarity of the image? Are there any areas where details are particularly sharp or blurry? Check for any loss of detail in specific regions. Provide specific examples of detail and resolution characteristics.
- "Are there any compression artifacts, sharpening artifacts, or color artifacts? Describe the location and intensity of these artifacts. Are there any signs of unnatural image processing? Identify any abnormal regions in the image and describe their characteristics. Provide specific examples of artifacts and anomalies.
- Does the image have a high or low dynamic range? Is the tone mapping natural, or does it appear unnatural? Check how well the highlights and shadows retain details. Are there any tone mapping anomalies, such as clipped highlights or crushed shadows? Provide specific examples of dynamic range and tone mapping characteristics.

**Limit**
Do not use relative terms like more, less, etc.

**Template**
Compare the low-level visual features of image_1 and image_2. Focus on the following aspects for each image: [{Perspectives[idx]}]. Format your response as follows: - image_1: [description], - image_2: [description]. But note: {limit}.

Figure C.1: The details of the prompt template employed within the low-level vision comparison pipeline for the full synthesis.

- Edges and contours.
- Brightness and contrast.
- Blurriness and sharpness.
- Color distribution.
- Distortions and anomalies.
- Noise and texture.
- Lighting and shadows.
- Level of detail and resolution.
- Artifacts and anomalies.
- Dynamic range and tone mapping.

Based on these ten perspectives, we designed corresponding prompt templates. The low-level vision comparison prompt for full synthesis is shown in Fig. C.1.

Specifically, for partial synthesis, since we only need to explore the low-level differences between the synthesized regions and their corresponding real counterparts, we incorporate region-specific nouns as parameters in the prompt. The low-level vision comparison prompt for partially synthesized images is illustrated in Fig. C.2.

## C.2   Analysis of Low-Level Vision Description

To mitigate survivor bias and potential discrepancies in low-level vision bias across different generators, for each category of full synthesis samples, we generated 12,000 differential descriptions from 10 distinct low-level vision perspectives, resulting in a total of 120,000 differential descriptions per full synthesis subset. Subsequently, we selected compound nouns from the 12,000 descriptions for each perspective and calculated their frequency proportions. In addition, it is necessary to generate corresponding real-image low-level vision comparison descriptions based on the synthesis sample categories and subsequently calculate the frequency proportions of compound nouns.

**10 Perspectives:**

- Describe the edges and contours in the {region} of the image in detail. Are the edges sharp, blurry, or fragmented? Are the outlines complete or irregular? Are there any areas with particularly high or low edge density in the {region}? Do the edges follow a specific direction (e.g., horizontal, vertical, diagonal)? Provide specific examples of edge and contour characteristics in the {region}.
- Analyze the brightness and contrast in the {region} of the image in detail. Is the {region} overall too bright, too dark, or well-balanced? Are there any areas with extreme highlights or shadows in the {region}? Describe the distribution of contrast across the {region}. Are there any signs of overexposure or underexposure in the {region}? Provide specific examples of brightness and contrast variations in the {region}.
- Evaluate the blurriness and sharpness of the {region} in the image in detail. Is the {region} overall sharp or blurry? If blurry, what type of blur is present (e.g., motion blur, Gaussian blur, defocus blur)? Are there specific areas within the {region} that are more blurred than others? Compare the sharpness of different parts of the {region}. Provide specific examples of sharp and blurry areas in the {region}.
- Describe the color distribution in the {region} of the image in detail. What are the dominant colors in the {region}, and how are they distributed? Are the colors evenly distributed in the {region}, or are there any anomalies like oversaturation, color distortion, or color blocks? Analyze the brightness and saturation of colors in the {region}. Provide specific examples of color characteristics and anomalies in the {region}.
- Check for any distortions or anomalies in the {region} of the image in detail. Are there any geometric distortions in the {region}, such as stretching, warping, or asymmetry? Are there any compression artifacts in the {region}, like blocky patterns or jagged edges? Analyze the symmetry of the {region}. Are there any unnatural deformations in the {region}? Provide specific examples of distortions and anomalies in the {region}.
- "Analyze the noise and texture in the {region} of the image in detail. Is the {region} noisy? If so, what type of noise is present (e.g., Gaussian noise, salt-and-pepper noise, quantization noise)? Describe the intensity and distribution of noise across the {region}. Analyze the texture complexity in the {region}—is it uniform or varied? Are there any directional patterns in the texture of the {region}? Provide specific examples of noise and texture characteristics in the {region}.
- Analyze the lighting and shadows in the {region} of the image in detail. Is the lighting in the {region} uniform or uneven? Are the shadows in the {region} natural and consistent with the lighting direction, or do they appear unnatural? Compare the highlights and shadows in different parts of the {region}. Are there any lighting anomalies in the {region}, such as overly bright or dark areas? Provide specific examples of lighting and shadow characteristics in the {region}.
- Evaluate the level of detail and resolution in the {region} of the image in detail. Are the details in the {region} rich and well-defined, or are they lacking? How does the resolution affect the clarity of the {region}? Are there any areas within the {region} where details are particularly sharp or blurry? Check for any loss of detail in specific parts of the {region}. Provide specific examples of detail and resolution characteristics in the {region}.
- Check for any artifacts or anomalies in the {region} of the image in detail. Are there any compression artifacts, sharpening artifacts, or color artifacts in the {region}? Describe the location and intensity of these artifacts in the {region}. Are there any signs of unnatural image processing in the {region}? Identify any abnormal regions within the {region} and describe their characteristics. Provide specific examples of artifacts and anomalies in the {region}.
- Analyze the dynamic range and tone mapping in the {region} of the image in detail. Does the {region} have a high or low dynamic range? Is the tone mapping in the {region} natural, or does it appear unnatural? Check how well the highlights and shadows in the {region} retain details. Are there any tone mapping anomalies in the {region}, such as clipped highlights or crushed shadows? Provide specific examples of dynamic range and tone mapping characteristics in the {region}.

**Limit**
Do not use relative terms like more, less, etc.

**Template**
Compare the low-level visual features of image_1 and image_2. Focus on the following aspects for each image: [{Perspectives[idx]}]. Format your response as follows: - image_1: [description], - image_2: [description]. But note: {limit}.

Figure C.2: The details of the prompt template employed within the low-level vision comparison pipeline for the partial synthesis.

**A simple example of the full synthesis samples in the SDXL category is illustrated in Fig. C.3.** This image illustrate the proportion of compound nouns at the low-level vision perspective for edges and contours, with the upper-left image displaying the noun proportion ranking for synthesis, and the lower-left image showing the noun proportion ranking for real.

To identify the nouns that may serve as judgment biases for MLLMs, we computed the maximum differences in noun proportions between real and synthesis images, with the results presented in the right half of Fig. C.3. It is evident that, compared to the SDXL category of synthesis images, real images exhibit a 20% proportion of 'smooth outlines', 'well-defined contours', and 'noticeable fragmentation', while over 70% display 'low edge densities'. In contrast, synthesis images have less than 20% exhibiting 'low edge densities, but a significantly higher proportion of 'high edge densities', alongside low-level visual features, such as 'irregular outlines' and 'lower edge density', not present in real images.

Therefore, for synthesis images in SDXL category, the judgment biases derived from the low-level vision analysis at the edges and contours perspective are as follows: they are likely to exhibit features such as irregular outlines and lower edge density, and, in comparison to real images, they are more prone to display high edge density.

**A simple example of the partial synthesis samples in the SDXL category is illustrated in Fig. C.4 and Fig. C.5.** Notably, for partial synthesis images, we generated differential descriptions from 10 distinct low-level vision perspectives for each of four selected regions, as outlined in Sec. B.1. For each category of partial synthesis images, a total of 480,000 differential descriptions were generated. We observed that even for partial synthesis derived from the same real, differences in the modified region led to slight differences in low-level vision bias. Therefore, in Fig. C.4 and Fig. C.5, we

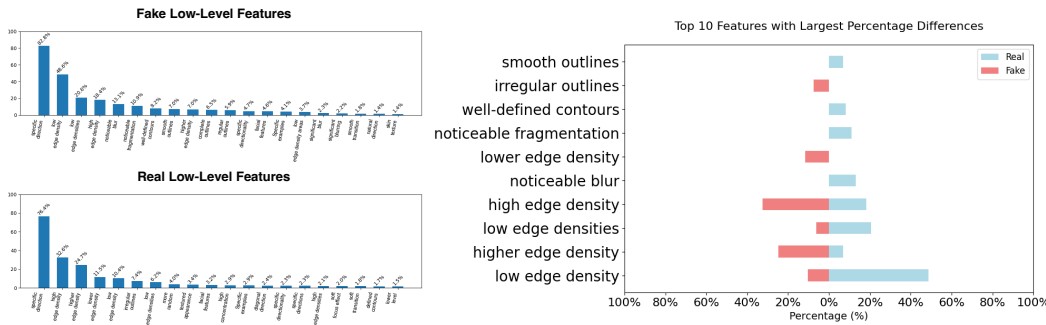

Figure C.3: The low-level vision bias obtained after processing full synthesis images through the low-level comparison pipeline.

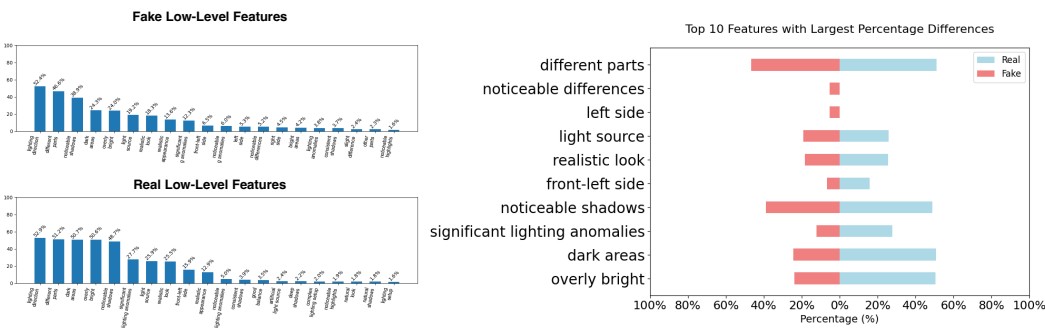

Figure C.4: The low-level vision bias obtained after processing hair-tampered images through the low-level comparison pipeline.

present the ranked data for two types of regions with significant size differences–hair (*i.e.*, Fig. C.4) and brows (*i.e.*, Fig. C.5) (low-level vision perspective: lighting and shadows).

It can be observed that even within the same low-level vision aspect, judgment bias varies significantly across different regions. For instance, in Fig. C.4, real images exhibit more pronounced characteristics in lighting and shadows perspective, such as 'overly bright areas', 'dark regions', and 'significant lighting anomalies' when compared to manipulated images with hair modifications. However, in Fig. C.5, real images compared to eyebrow-edited fake images display 'higher contrast', a 'more natural appearance', and 'multiple lighting angles'—features that are not observed in the manipulated images.

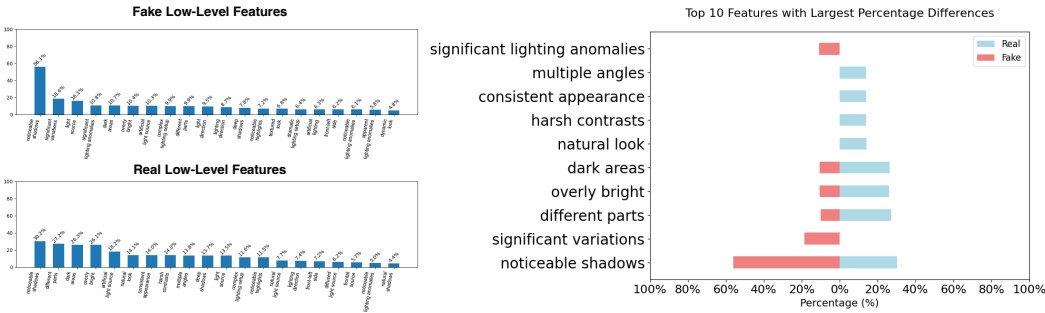

Figure C.5: The low-level vision bias obtained after processing brow-tampered images through the low-level comparison pipeline.

## C.3 Analysis of Low-Level Vision Bias

During the noun selection process, issues such as synonymy, ambiguous terms, and minimal proportions differences in noun characteristics between real and fake images often arise. Consequently, not every feature with a large proportional difference should be considered a judgment bias. For example, 'low edge density' and 'low edge densities' in Fig. C.3, as well as 'different parts' and 'left side' in Fig. C.4, illustrate such cases. Therefore, manual filtering is required to identify distinguishing low-level vision features that can effectively differentiate real from fake images.

In Fig. C.6, we demonstrate the process of applying the low-level vision comparison pipeline to a fake image generated by SDXL. The pipeline extracts low-level vision biases, which are then manually filtered to obtain the final judgment bias descriptors.

# D Full Experiment Result of VLF

This section provides a detailed description of all aspects involved in the experimental phase of the VLForgery framework, including the quantification of textual outputs, error analysis, additional metric evaluations, and extended experiments.

## D.1 Quantitative Analysis of Results

| Name | Train–SDXL%↓ | | SD2%↓ | | Kandinsky2.2%↓ | | SD3%↓ | Flux%↓ | InstantID%↓ | Avg%↓ |
|---|---|---|---|---|---|---|---|---|---|---|
| | PS | FS | PS | FS | PS | FS | | | | |
| VariantA | 0.00 | 0.00 | 0.00 | 0.00 | 0.00 | 0.00 | 0.00 | 0.00 | 0.00 | 0.00 |
| VariantB | 0.00 | 3.52 | 0.00 | 3.11 | 0.00 | 3.88 | 1.60 | 5.83 | 6.45 | 2.71 |

Table D.1: Error study.

**Result quantification:** We employed regular expressions to filter MLLM outputs across multiple tasks. During testing, while the answers seemed correct, they did not always match the groundtruth due to issues like case mismatches (*e.g.*, 'True' vs. 'true', 'Fake' vs. 'fake'). Although synonyms did occur (*e.g.*, 'fake' vs. 'false'), such cases were extremely rare. To address this, we created a synonym pool and preprocessed the results. If the model's output matched an entry in the synonym pool, we considered it correct (*e.g.*, 'false' was treated as 'fake').

**Error Analysis:** To address the inconsistency in detection results, we analyzed two key issues: 1. Variant A: Mismatch between authenticity judgment and forgery type (e.g., detected as 'real' but labeled as 'full synthesis'). 2. Variant B: Mismatch between forgery type and falsified region (e.g., labeled as 'full synthesis' but falsified region is 'hair'). The results are shown in Tab D.1.

## D.2 Experimental Results Supplement

**Comparative Analysis of Single and Multi-edited Images.** Given that multi-edited faces may disturb the detector's judgment, we divided the test data for each region into single-edited and multi-edited groups. We fine-tuned VLForgery on SDXL partial synthesis and subsequently tested it on both single-edited and multi-edited faces across four regions' test sets. As shown in Fig. D.1, we visualized the localization results of these two groups for four regions, respectively.

The results reveal that, although localization difficulty varies across regions, single-edit images are generally easier to locate compared to multi-edit images.

**Additional metrics for evaluation.** We introduce two additional metrics, F1 and Precision, to comprehensively evaluate the detection performance of different models. Tab. D.2 illustrates the precision performance of different models across various subset test sets, while Tab. D.3 demonstrates their F1 performance on the same subsets.

# E Additional Information on VLForgery

In this section, we present all additional details related to the VLForgery framework that were not covered in previous sections. These include the prompt design, question formulation, and inference result examples used in the fine-tuning and inference modules of MLLMs; the full low-level vision

**Top 10 Features with Largest Percentage Differences**

**Artifacts:**
- Most fake images lack visible compression artifacts, sharpening artifacts, or color artifacts, while real images may occasionally show these.
- Fake images often exhibit higher image quality.
- A small number of fake images may show unnatural semantic signs.

**Blurriness and Sharpness:**
- Most real and fake images show little visible blur.
- Some fake images may still have blurry areas.
- Real images often have clearer and more detailed skin textures compared to fake ones.
- A few fake images may show unfocused blur.

**Brightness and Contrast:**
- Most real images have balanced brightness and contrast, while some fake images may show imbalances.
- Real images usually have neutral overall tones, while some fake images may have more varied tones.
- Most images have balanced exposure, but some fake images may show uneven exposure.

*Color Distribution:**
- Both real and fake images generally have uniform color distribution, but fake images may be slightly less uniform.
- A very small number of fake images may have a dominant earthy tone, which is rare in real images.
- Real images may show soft glows around subjects, which is uncommon in fake images.
- Real images are more likely to have oversaturated colors or natural tones compared to fake images.

**Distortions and Anomalies:**
- Most real images are natural close-up portraits under natural light.
- Real images often have features captured at low resolution, which is rare in fake images.

Figure C.6: The process of manually extracting final judgment bias descriptors from low-level vision bias

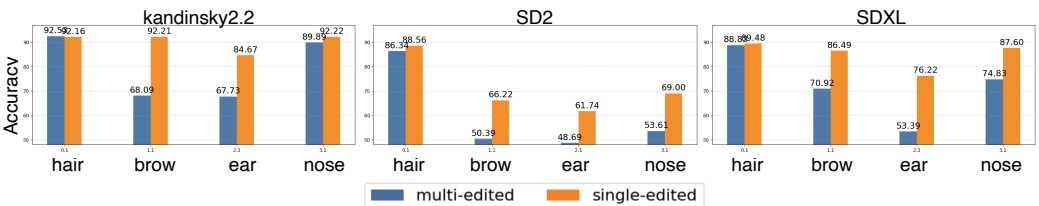

Figure D.1: Visualization of model's localization accuracy on single-edited and multi-edited faces. The Large ACC area means better localization capability.

| Method | Train–SDXL↑ | | SD2↑ | | Kandinsky2.2↑ | | SD3↑ | Flux↑ | InstantID↑ | Avg↑ |
|---|---|---|---|---|---|---|---|---|---|---|
| | PS | FS | PS | FS | PS | FS | | | | |
| Resnet-50 | 90.77 | 89.82 | 76.88 | 89.43 | 94.24 | 89.6 | 88.9 | 89.27 | 77.71 | 87.4 |
| Xception | 79.83 | 64.69 | 69.94 | 64.14 | 84.9 | 64.62 | 64.17 | 64.47 | 43.19 | 66.66 |
| SPSL | 90.1 | 90.08 | 57.95 | 89.8 | 93.67 | 90.07 | 89.8 | 90.02 | 78.26 | 85.53 |
| DRCT* | 86.7 | 78.02 | 86.87 | 77.73 | 91.83 | 78.11 | 70.2 | 67.84 | 59.55 | 77.38 |
| NPR | 65.52 | 58.43 | 31.27 | 58.19 | 75.84 | 58.33 | 58.01 | 58.38 | 36.8 | 55.64 |
| CLIPping *Adapter* | 53.45 | 49.96 | 50.42 | 48.54 | 70.37 | 49.58 | 49.63 | 49.47 | 28.78 | 50.02 |
| CLIPping *LP* | 66.4 | 50.25 | 63.48 | 49.54 | 73.64 | 50.16 | 49.72 | 50.01 | 29.58 | 53.64 |
| SRM | 69.34 | 54.66 | 31.96 | 54.33 | 78.97 | 54.56 | 54.28 | 54.21 | 32.78 | 53.9 |
| F3Net | 91.40 | 81.91 | 87.08 | 81.84 | 92.14 | 81.91 | 81.82 | 81.92 | 65.34 | 82.81 |
| GramNet | 87.42 | 86.52 | 74.38 | 86.16 | 90.77 | 86.48 | 85.71 | 86.44 | 70.29 | 83.80 |
| SAFE | 61.92 | 42.73 | 60.89 | 42.58 | 61.85 | 41.8 | 44.57 | 34.15 | 21.3 | 45.75 |
| CNNspot | 91.4 | 92.59 | 78.5 | 90.3 | 91.59 | 90.43 | 83.59 | 89.93 | 79.64 | 87.55 |
| VLForgery | 91.41 | 81.04 | 89.98 | 81.03 | 93.12 | 81.03 | 81.04 | 81.03 | 63.94 | 82.62 |

Table D.2: Precision.

| Method | Train–SDXL↑ | | SD2↑ | | Kandinsky2.2↑ | | SD3↑ | Flux↑ | InstantID↑ | Avg↑ |
|---|---|---|---|---|---|---|---|---|---|---|
| | PS | FS | PS | FS | PS | FS | | | | |
| Resnet-50 | 50.61 | 94.52 | 20.56 | 92.45 | 72.11 | 93.32 | 89.73 | 91.59 | 85.32 | 76.69 |
| Xception | 73.55 | 78.56 | 50.98 | 77.4 | 90.5 | 78.41 | 77.47 | 78.1 | 60.25 | 73.91 |
| SPSL | 46.81 | 94.77 | 8.85 | 93.22 | 66.39 | 94.7 | 93.22 | 94.42 | 85.86 | 75.36 |
| DRCT* | 69.25 | 87.43 | 69.93 | 86.58 | 95.43 | 87.68 | 68.02 | 63.15 | 74.34 | 77.98 |
| NPR | 51.68 | 73.76 | 15.41 | 73.29 | 73.08 | 73.57 | 72.95 | 73.65 | 53.71 | 62.34 |
| CLIPping *Adapter* | 43.23 | 66.61 | 39.25 | 64.1 | 72.63 | 65.93 | 66.02 | 65.73 | 44.4 | 58.66 |
| CLIPping *LP* | 64.01 | 66.88 | 58.55 | 65.62 | 79.9 | 66.73 | 65.94 | 66.46 | 45.63 | 64.41 |
| SRM | 63.87 | 70.67 | 17.76 | 70.07 | 87.6 | 70.5 | 69.97 | 69.85 | 49.01 | 63.26 |
| F3Net | 81.84 | 90.02 | 61.02 | 89.79 | 86.58 | 90.01 | 89.73 | 90.05 | 78.99 | 84.23 |
| GramNet | 49.13 | 92.77 | 23.96 | 91.23 | 63.11 | 92.6 | 89.4 | 92.41 | 78.33 | 74.77 |
| SAFE | 59.07 | 56.21 | 57.28 | 55.94 | 58.96 | 54.66 | 59.27 | 42.73 | 32.81 | 52.99 |
| CNNspot | 40.85 | 95.18 | 16.21 | 80.73 | 41.67 | 81.47 | 54.06 | 78.72 | 76.5 | 62.82 |
| VLForgery | 84.53 | 89.52 | 76.36 | 89.51 | 96.42 | 89.51 | 89.52 | 89.51 | 77.87 | 86.97 |

Table D.3: F1.

bias results from the low-level vision comparison pipeline; as well as comparative examples of inference results from other multimodal language models.

## E.1 Details of the Fine-Tuning and Inference Modules

This section describes the fine-tuning and inference module and the various templates required for EkCot generation.

Fig. E.1 outlines the specific contents of the general rule template involved in the description generation module. The templates incorporate additional knowledge, such as ground truth and low-level vision bias, represented by specific placeholders. For example, 'detection_result' indicates whether the image is real or fake, 'text_prompt' represents the prompt used during the image generation process, and 'judgmentbias' denotes the low-level vision judgment bias knowledge required for analysis.

Fig. E.2 illustrates all the question formats randomly generated by GPT-4o for the final fine-tuning dataset.

## E.2 Comparison of the Inference Result with Other MLLMs

We randomly selected three representative samples of forged images, each corresponding to a distinct type of forgery, and demonstrated the reasoning outcomes generated by our framework in comparison to those produced by other open-source Multimodal Large Language Models (MLLMs). Fig. E.3 presents the analysis results generated by different reasoning models for a synthetic image created through identity insertion. Fig. E.4 presents the analysis results generated by different reasoning models for a synthetic image created through sdxl model. Fig. E.5 presents the analysis results

You are an expert in detecting and analyzing face forgeries.

We categorize detection tasks into three primary aspects:
        1.Detecting the authenticity of faces.
        2.Identifying falsified regions.
        3.Tracing the generation process of forgery faces.
We also categorize forgery datasets into five types of forgerys
        1.Facial fine-grained semantic editing.
        2.Identity transformation.
        3.Non-existent faces.
You will be provided with the following informations:
        - A text prompt used during the generation process of the image: {text_prompt}
        - The detection result of the image: {detection_result}
        - The forgery type of the image: {forgery_type}
        - The falsified region of the image: {falsified_region}
        - The method used to generate the forgery: {forgery_method}

I need your help to determine whether an image is AI-generated or real. I have provide you with
an image and tell you all the groundtruth and informations about this image, but I am unsure what
specific features make it fake or real . Based on my research, here are some key differences
between real and AI-generated images that you should analyze:{judgmentbias}

**Output format:**
1.**Detection Result**: real/fake.
2.**Forgery analysis**: Based on the key differences I have provided to you. analyze the image I
provide and explain why it might be fake or real. Focus on identifying specific features or
anomalies that align with the characteristics of AI-generated images as described above. Provide a
detailed explanation of your findings.
3.**Forgery Type**: facial fine-grained semantic editing/identity transformation/non-existent
faces/none.
4.**Falsified Region**: hair/ear/nose/brow/....../full synthesis/none.
5.**Forgery Method**:
SDXL/SD2/Kandinsky/Flux/Midjourney....../Playground/Titan/Ideogram/none.

**Rules**:
        1. Do not use bold for any titles in the output.
        2. Generate answers strictly within the word limit.

Figure E.1: Specific details of the General Rule Template used in the forgery description module.

generated by different reasoning models for a partial synthetic image created through sdxl-inpainting
model.

- Does this image show a genuine or artificially created human face? What clues help you decide?
- Is the human face depicted in this image real or generated? What evidence supports your answer?
- Do you believe this face in the image is natural or synthetically created? What leads you to that conclusion?
- Looking at this face, is it an authentic human face or a digital imitation? What observations influence your judgment?
- Can you tell if the face in this image is unmodified or manipulated? What details did you consider?
- Is this a true representation of a human face or a manipulated one? Explain your reasoning.
- Would you say this image contains a real human face or an artificial one? What factors helped you determine that?
- This image features a human face. Is it authentic or digitally created? What makes you think so?
- Do you think the face shown in this image is genuine or doctored? What clues did you find?
- Is the human face in this image unedited or altered? What details support your conclusion?
- Does this image present a real human face or a computer-generated one? What observations led you to that assessment?
- In this image, is the human face natural or modified? What makes you draw that conclusion?
- Would you classify the face in this image as real or synthetic? What evidence do you base your answer on?
- Is this an actual human face or a digitally constructed one? Describe the signs that influenced your evaluation.
- Does this face look like a natural one, or does it appear to be digitally altered? What indicators made you think that?
- Can you determine if the human face here is authentic or fabricated? What details did you analyze?
- Is the human face shown genuine, or does it have signs of digital editing? What observations led you to your conclusion?
- Does this image feature a true human face or a manipulated one? What evidence supports your interpretation?
- Do you think the face in this image is original or has been digitally adjusted? Explain your observations.
- Is the human face depicted here real or a product of digital manipulation? What clues indicate your choice?
- The image contains a human face. Would you say it's unedited or modified? What factors influenced your judgment?
- Is this a faithful depiction of a human face, or does it show signs of alteration? What details influenced your view?
- Is the face in this image authentic or tampered with? What observations led you to your conclusion?
- Does this appear to be a genuine human face or one that's digitally fabricated? What cues influenced your perspective?
- Would you say this image represents a natural human face, or does it seem artificial? What details did you consider?
- Is the human face in this image untouched or altered? What leads you to your judgment?
- The image depicts a human face. Do you believe it's authentic or digitally altered? Explain your reasoning.
- Is this a real human face, or has it been digitally altered? What makes you think that?
- Looking at this image. Is it genuine or a fake? What clues lead you to your conclusion?
- Do you believe this face looks authentic or manipulated? What reasoning backs up your belief?
- Does the image depict a real human face, or does it show evidence of editing? What clues suggest this to you?
- Can you identify whether this face is naturally occurring or generated? What led you to that decision?
- Does this image show an unmodified human face or one that's digitally constructed? What observations support your claim?
- Is the face in this image genuine, or does it appear to be digitally produced? What makes you think so?
- Do you see this as a natural human face or a synthetic one? What evidence backs up your impression?
- Does the human face in this image seem real, or does it give signs of digital creation? What led you to conclude that?
- Would you consider this human face real or manipulated? What details in the image suggest this to you?
- Does the human face look natural or altered? What observations support your view?
- Do you believe the image presents a real human face, or does it have features of digital editing? What clues support this?
- This is a image of a human face. Can you determine if it's real or artificial? What supports your answer?
- Does the image contain a natural human face or one that is artificially generated? What leads you to believe that?
- Is this a digitally created or a real human face? What evidence do you rely on?
- Does the face shown in the image appear genuine or manipulated? What factors influence your decision?
- Can you determine if this human face is real or digitally modified? What indicators did you examine?
- A human face appears in the image. Do you think it's real or altered? What makes you say so?
- There's a human face in this image. Is it natural or fake? What evidence supports your conclusion?
- This image displays a human face. Is it genuine or manipulated? What leads you to that decision?
- A human face is present in this image. Is it authentic or doctored? Provide your reasoning.
- This image shows a human face. Do you think it's real or a forgery? What observations support your answer?

Figure E.2: Randomly generated question descriptions using ChatGPT-4o.

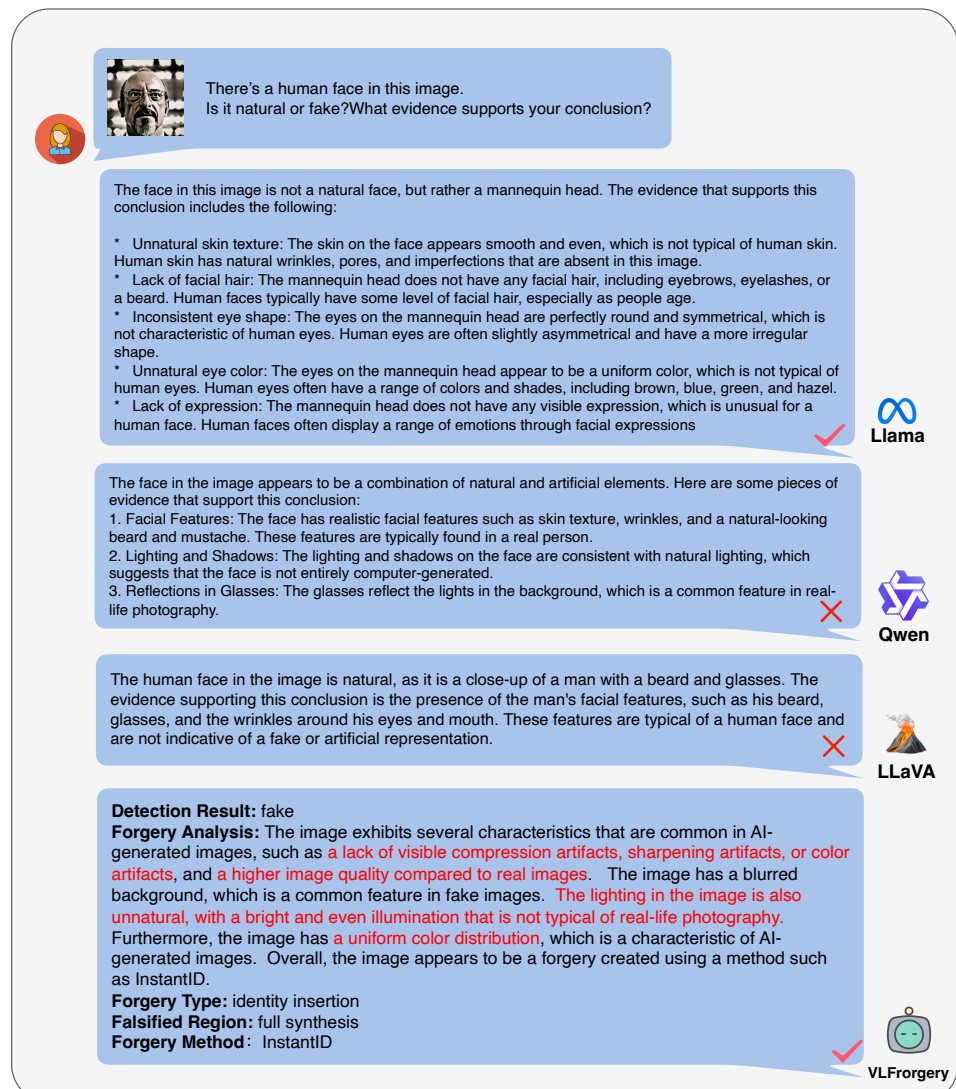

Figure E.3: Comparison of responses to the full synthesis between other widely used MLLMs and the VLForgery framework. The sample shown is an identity insertion face generated by the InstantID model.

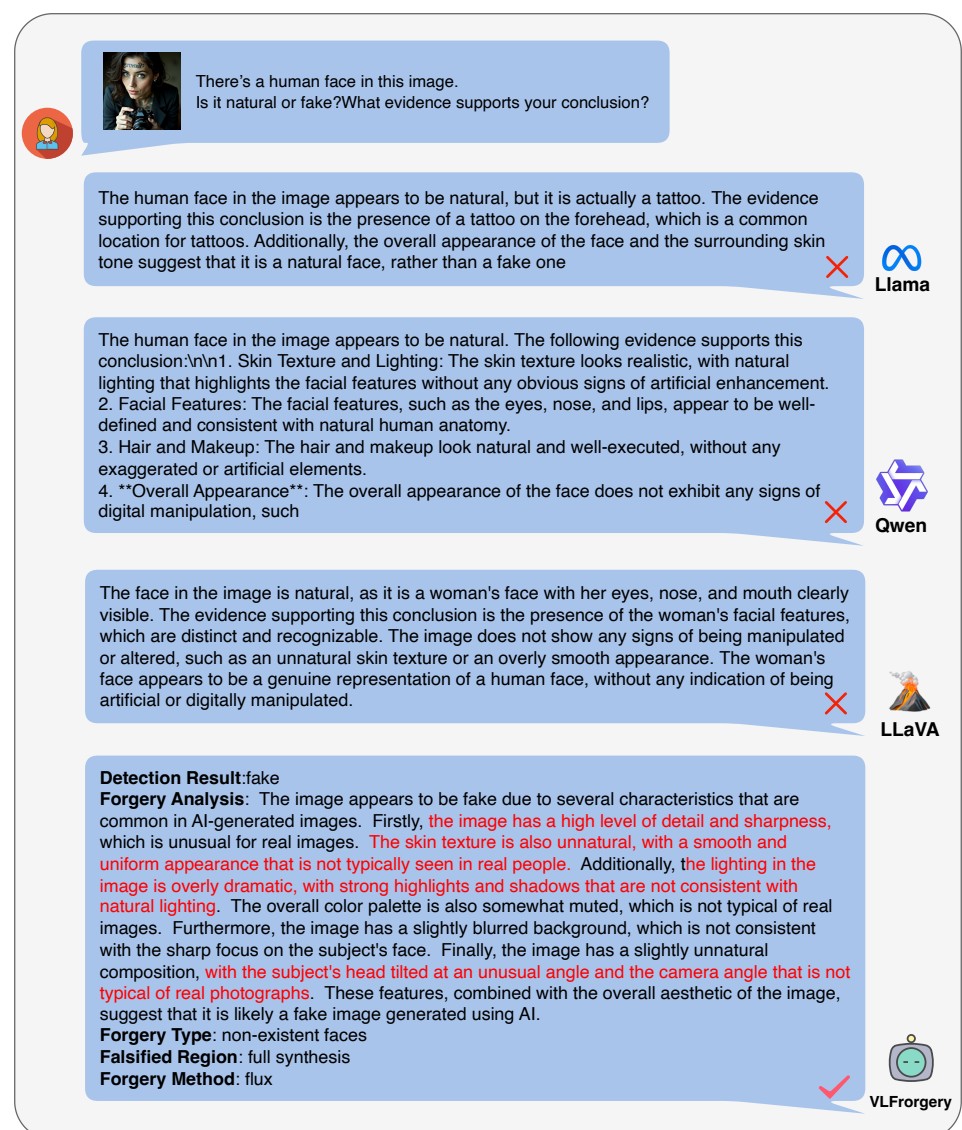

Figure E.4: Comparison of responses to the full synthesis between other widely used MLLMs and the VLForgery framework. The sample shown is a non-existent face generated by the flux model.

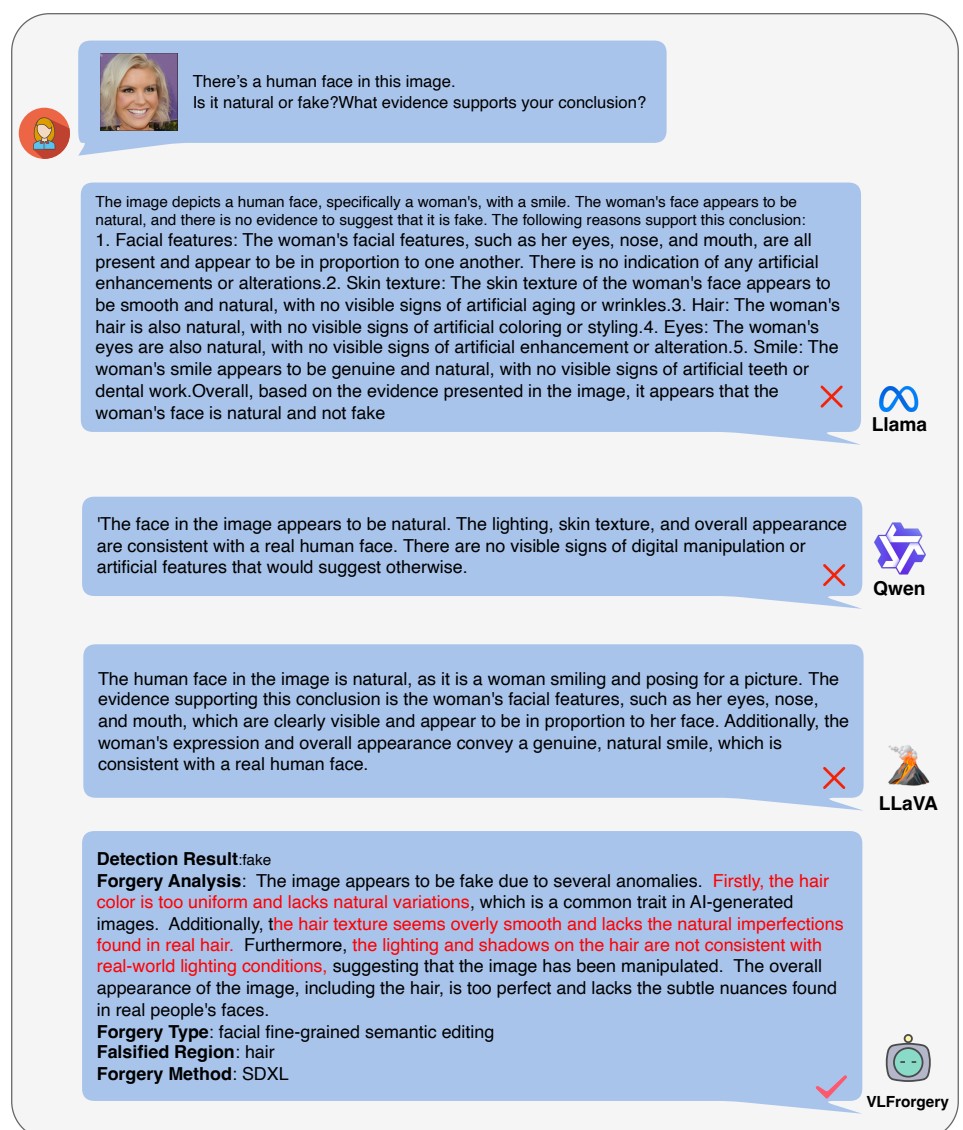

Figure E.5: Comparison of responses to the partial synthesis between other widely used MLLMs and the VLForgery framework. The sample shown is a hair-edited face generated by the sdxl model.

