# OpenReview forum: "VLForgery Face Triad: Detection, Localization and Attribution via Multimodal Large Language Models"
_NeurIPS.cc/2025/Conference — NeurIPS 2025 poster_

### Official Review · Reviewer_fi2j · 2025-06-30

**Clarity:** 3
**Significance:** 3
**Originality:** 3
**Rating:** 4
**Confidence:** 3

**Summary:**

Images generated by the diffusion models pose a significant challenge for Deepfake detection. This paper proposed a Multimodal Large Language Model (MLLM) based deepfake detection framework named VLForgery. This framework identifies manipulated regions and distinguishes different synthetic generators, thereby effectively detecting forgeries. To this end, the paper first collected the VLF (Visual Language Forensics) dataset with various generators. Then, the Extrinsic knowledge-guided Chain-of-thought (EkCot) is constructed to guide the MLLM to detect forgeries. Finally, the paper proposes a low-level vision comparison pipeline to explore the significant differential features between real and fake images. Extensive experiments demonstrate the superior performance of the proposed method.

**Questions:**

Please refer to the weakness.

**Ethical Concerns:**

["NO or VERY MINOR ethics concerns only"]

**Final Justification:**

Thank you to the authors for their response and the additional experiments. It addresses my concern. I maintain my recommendation of borderline accept.

**Limitations:**

yes

**Paper Formatting Concerns:**

The reviewer has not found any formatting issues in the paper so far.

**Quality:**

3

**Strengths And Weaknesses:**

**Strengths**

1. The Extrinsic knowledge-guided Chain-of-thought (EkCot) improved the interpretability of large models for forgery detection. The method provides new insight into the explainability of deepfake detection based on multimodal large language models.
2. The paper conducts a comprehensive evaluation experiment to demonstrate the superior performance of the proposed method.

**Weaknesses**

1. In this paper, the VLF dataset generates fake images with published manipulation methods. The data collection pipeline is similar to previous work [1]. It seems that the paper should discuss the difference between VLF and other manipulation detection datasets to demonstrate the contribution of the proposed benchmark.
2. In Table 4, the vanilla binary classifier based on Xception achieved high forgery detection accuracy on cross-dataset evaluation. Such experimental results indicate a strong correlation among forgeries generated by different manipulation methods within the VLF dataset. The paper should conduct cross-dataset evaluation experiments on more challenging benchmarks, such as DFDC [2] and FF++ [3].
3. In this paper, the VLForgery is only compared with a few vanilla binary classifiers. More recent works on deepfake detection, such as [4, 5, 6], should be considered in the experiments.
4. Robustness toward different kinds of perturbations is also an important part of DeepFake detection. The paper should conduct such experiments to indicate the robustness of the proposed method.

[1] Diffusion Facial Forgery Detection

[2] The deepfake detection challenge dataset.

[3] Faceforensics++: Learning to detect manipulated facial images.

[4] Face X-Ray for More General Face Forgery Detection

[5] Detecting Deepfakes With Self-Blended Images

[6] Implicit Identity Leakage: The Stumbling Block to Improving Deepfake Detection Generalization

---

> ### Author Rebuttal · Authors · 2025-07-30
>
> ## Response to Weakness 1: Differences Between VLF and Prior Manipulation Detection Datasets
>
> Thank you for this important comment. At the early stage of our work, we indeed conducted a careful survey of existing datasets such as [1] and others. However, we found that they could not fully meet the needs of our task. Below, we outline key differences and contributions of the proposed VLF benchmark:
>
> 1. **Prompt Design for Face Generation**:
>    Existing AIGC datasets (e.g., [1], [2]) often use short and simplistic prompts, which may not produce low-quality images, but result in less detailed and diverse facial features.
>    In contrast, we design long-format prompts that are*specifically tailored for face generation, inspired by dedicated YouTube channels studying facial synthesis (details in Section B.3). Our prompts include semantically rich tokens such as `[framing]`, `[camera_angle]`, `[lighting]`, etc., which significantly enhance facial realism and diversity.
>
> 2. **Generator Quality and Coverage**:
>    [1] often rely on privately trained or lower-quality open-source models, which may not match the realism of widely-used diffusion models like Flux, Kandinsky, etc.
>
> 3. **Detailed Annotations for Partial Synthesis**:
>    Our task focuses on fine-grained forgery localization and attribution, which requires precise metadata. Existing datasets rarely provide such details.
>    In VLF, we include annotations on:
>    - Whether edits involve single-region or multi-region manipulation;
>    - The exact name and location of modified regions;
>    - Whether the image was synthesized from scratch or by editing real images.
>
> - [1] Diffusion Facial Forgery Detection
> - [2] GenImage: A Million-Scale Benchmark for Detecting AI-Generated Image
>
> ## Response to weakness 2: Cross-Dataset Evaluation on Challenging Benchmarks (e.g., DFDC, FF++)
>
> We appreciate the reviewer’s suggestion regarding broader cross-dataset evaluation. We agree that the current version of our framework has not been extensively evaluated on traditional Deepfake datasets, which could limit the generalizability of our conclusions.
>
> Due to time and computational resource constraints, we were unable to build and run a dedicated training/testing pipeline on large-scale Deepfake datasets like DFDC. However, as an initial step, we conducted evaluation on the face-swapping subset of FaceForensics++.
> | Train | DF    | F2F   | FS    | FST   | NT    | Avg(%)↑ |
> |-------|-------|-------|-------|-------|-------|---------|
> | DF    | 96.11 | 96.34 | 95.74 | 95.98 | 95.90 | 96.01   |
> | F2F   | 98.79 | 99.09 | 98.63 | 98.38 | 98.47 | 98.67   |
> | FS    | 99.05 | 99.20 | 99.05 | 98.99 | 99.05 | 99.07   |
> | FST   | 97.09 | 97.30 | 96.57 | 97.01 | 97.01 | 96.99   |
> | NT    | 99.72 | 99.70 | 99.56 | 99.60 | 99.62 | 99.56 |
>
> *Table: Evaluation on Traditional Face-swapping Deepfakes type. Results on different training and testing subsets using VLForgery. Accuracy is used for evaluation.*
>
> We will include these results in the revised version of the paper and plan to expand to additional benchmarks in future work.
>
> ## Response to Weakness 3: Limited Baselines – Consideration of Recent Deepfake Detection Methods [4, 5, 6]
>
> Thank you for this valuable comment. We understand your concern regarding the comparison with only a few baseline methods. Below, we clarify our rationale:
>
> Although both traditional Deepfake datasets (e.g., [FF++]) and AIGC-based datasets (e.g., [1]) focus on face manipulation, **the manipulation mechanisms differ fundamentally**. Traditional Deepfakes are typically generated via **face-swapping or reenactment techniques** such as StarGAN, FaceAPP, or FaceSwap, whereas our dataset, VLForgery, is based on **diffusion models** and **prompt-driven generation**.
>
> Methods like [4, 5, 6] enhance forgery detection by exploiting **artifacts specific to face-swapping** (e.g., warping, boundary inconsistencies). While these are effective for traditional Deepfakes, such artifacts **do not exist** in AIGC-based forgeries. Instead, recent works for AIGC forensics (e.g., DRCT using self-reconstruction, SAFE and CNNspot using traditional image augmentation, and SRM, SPSL, NPR employing frequency-based features) are more suitable.
>
> Therefore, we believe our inclusion of these **AIGC-relevant forensics methods**—particularly those utilizing frequency priors—is more aligned with the nature of our benchmark. That said, we acknowledge that our current baselines might appear limited, and we are open to incorporating more recent models in future versions of the paper. We sincerely welcome further discussion with the reviewer to reach a consensus.
>
> ## Response to Weakness 4: Robustness Against Perturbations
>
> We agree that robustness toward different types of perturbations is a crucial aspect of DeepFake detection. To investigate this, we conducted robustness experiments using five commonly used perturbation types:
>
> - **GB**: Gaussian Blur
> - **BW**: Block-wise Masking
> - **CC**: Color Contrast
> - **CS**: Color Saturation
> - **JC**: JPEG Compression
>
> We selected representative detectors from three categories, including our proposed VLForgery. For simplicity and to reduce resource cost, we applied a single fixed severity level for each perturbation type. The detailed results are shown in the table below.
>
> |Robustness Type|||||||
> |--|--|--|--|--|--|--|
> |**Train-SDXL（PS)**|f3net|gramnet|srm|CLIPing_linear|VLF|
> |GB|69.63|37.12|25.17|55.68|68.23
> |BW|72.89|32.86|60.74|52.19|69.32
> |CC|72.91|37.89|52.91|58.33|75.83
> |CS|71.92|32.79|52.17|48.81|58.64
> |JC|72.30|25.10|3.47|29.11|59.91
> |**Trani-SDXL(FS)**|
> |GB|43.00|99.92|99.35|99.98|99.67
> |BW|37.68|99.95|99.98|99.98|99.87
> |CC|41.90|97.93|87.97|99.98|99.54
> |CS|41.22|99.73|99.43|99.98|99.98
> |JC|42.28|99.65|72.42|99.22|99.31
> |**SD2(PS)**|
> |GB|45.12|32.03|0.11|51.57|67.34
> |BW|45.47|23.79|14.98|50.42|68.19
> |CC|44.29|28.61|32.63|47.63|65.43
> |CS|43.00|20.37|26.86|40.12|63.24
> |JC|43.03|7.54|0.15|45.43|60.17
> |**SD2(FS)**|
> |GB|32.67|98.48|96.30|99.75|98.73
> |BW|25.83|96.30|99.02|97.33|98.62
> |CC|32.20|93.47|89.00|96.08|94.31
> |CS|31.18|95.83|97.50|96.70|97.62
> |JC|31.82|83.17|85.73|93.93|92.89
> |**Kan(PS)**|
> |GB|65.20|56.07|2.34|79.73|87.35
> |BW|72.31|56.10|98.87|70.23|89.25
> |CC|75.78|52.27|77.74|75.99|87.31
> |CS|74.06|49.17|81.91|71.36|70.94
> |JC|73.79|33.08|1.64|32.57|65.45
> |**Kan(FS)**|
> |GB|38.95|99.65|89.15|99.98|99.92
> |BW|32.32|99.53|99.62|99.58|99.87
> |CC|38.87|98.53|87.05|99.60|98.64
> |CS|36.93|98.85|98.98|99.50|99.17
> |JC|38.18|98.43|85.97|99.23|99.33
> |**SD3**|
> |GB|29.96|95.17|93.45|99.52|98.35
> |BW|27.97|91.82|98.35|98.05|97.83
> |CC|28.62|90.57|84.12|98.17|99.23
> |CS|27.75|90.50|96.32|95.48|93.27
> |JC|29.02|88.92|64.98|72.72|89.13
> |**Flux**|
> |GB|35.50|99.13|94.02|99.85|99.82
> |BW|32.28|99.10|98.30|99.13|99.09
> |CC|36.85|98.08|85.72|99.25|99.14
> |CS|33.23|98.63|95.55|98.42|99.03
> |JC|34.05|98.18|74.77|87.53|88.27
> |**InstantID**|
> |GB|63.34|98.66|90.33|100.00|99.48
> |BW|67.74|98.94|97.41|99.76|99.39
> |CC|72.81|97.45|81.93|99.61|99.41
> |CS|69.43|98.59|92.73|99.69|98.39
> |JC|68.06|97.80|49.39|96.66|97.82
>
>
> Key observations include:
>
> - **JPEG compression (JC)** tends to significantly affect most detectors.
> - There is noticeable variation in robustness across detectors and test sets.
> - Our method demonstrates a relatively balanced robustness profile — unlike SRM, which fails entirely under JC, or F3Net, which collapses in performance on fully synthesized samples.

---

> > ### Comment · Reviewer_fi2j · 2025-08-05
> > **Official Comment by Reviewer fi2j**
> >
> > Thank you to the authors for their response and the additional experiments. It addresses my concern. I maintain my recommendation of borderline accept.

---

> > > ### Author Response · Authors · 2025-08-06
> > >
> > > We truly appreciate your thoughtful feedback and suggestions, which have been very helpful in improving our work.

---

### Official Review · Reviewer_9dbu · 2025-07-02

**Clarity:** 3
**Significance:** 4
**Originality:** 3
**Rating:** 4
**Confidence:** 5

**Summary:**

This paper introduces a framework for forgery face detection, localization, and attribution based on MLLM, with the ability to perform fine-grained analysis. Its primary goal is to tackle the challenges of detecting Deepfakes created by high-fidelity human faces generated through diffusion models. Given that existing public datasets lack detailed labels, the authors present the VLF dataset and employ GPT4o alongside various diffusion models to generate forged face data. Additionally, the paper introduces the EkCot (Extrinsic knowledge-guided Chain-of-thought) method, which combines external knowledge from the image generation pipeline with low-level visual differences to guide MLLMs in performing credible forgery analysis. Through extensive experiments, the authors demonstrate that VLForgery outperforms current methods in detection accuracy and shows promising results in localization and attribution tasks.

**Questions:**

1. Robustness to MLLM hallucinations: Although EkCot aims to reduce the hallucination problem in MLLMs, how well does this mechanism hold up when faced with entirely new or very subtle forgery traces? The paper mentions that MLLMs lack fine-grained forensic awareness, which can lead to hallucinations—how does EkCot fundamentally address this issue?
2. The results in Table 6 suggest that VLForgery has much lower attribution accuracy for most forgery methods in the Full Synthesis category, as well as for Partial Synthesis type Kan2.2, compared to traditional methods. It only outperforms in three other forgery methods. Could you provide more detailed analysis and explanation of the underlying reasons for this?
3. Additionally, has the proposed method been tested for robustness against Unseen Perturbations (such as Gaussian blur, JPEG compression, etc.)?

**Ethical Concerns:**

["NO or VERY MINOR ethics concerns only"]

**Final Justification:**

The authors have fully explained and addressed the concerns I raised in the rebuttal. This work provides valuable insights for future research on deepfake detection using MLLMs. I maintain my original rating.

**Limitations:**

yes

**Quality:**

3

**Strengths And Weaknesses:**

### Strengths
1. Most existing forgery face/deepfake detection techniques are either classification models or a combination of classification and segmentation models. This work stands out by integrating all three tasks into a single MLLM-based framework, validating its effectiveness through experiments. This provides a more comprehensive tool for analyzing high-fidelity AI-generated content and its forgeries.

2. The VLF dataset, which is carefully crafted and designed, is a notable contribution to the field. It includes diverse types of forgeries (both partial and full synthesis) and various generators (SDXL, SD2, Kandinsky, SD3, Flux, InstantID), significantly filling the gap left by current datasets that lack fine-grained support for tasks like localization and attribution. This dataset provides valuable resources for future research.

### Weaknesses
1. Generalization issues for fully synthesized images across subtypes: As shown in Table 2, within the fully synthesized category, the model experiences a significant drop in accuracy under certain cross-type test scenarios (e.g., training on SD3 or Flux but testing on InstantID, where accuracy drops to 21.93% and 48.1%, respectively). This suggests that the model may struggle to generalize when confronted with generators that differ substantially, even within the fully synthesized category, and may fail to capture universal generator fingerprints.

2. While the VLF dataset, along with the low-level visual comparison pipeline and EkCot, enhances the model's perception of fine-grained features, the experiments reveal that the model still faces performance bottlenecks when dealing with local synthesis, especially in smaller regions or cross-generator cases (as shown in Tables 5 and 6). This could indicate that the current low-level visual feature extraction or EkCot method may need further refinement to better capture and utilize these subtle local differences within MLLM.

---

> ### Author Rebuttal · Authors · 2025-07-30
>
> ## Response to Weaknesses 1: Generalization Issues Across Fully Synthesized Subtypes (e.g., InstantID)
>
> We also noticed that **performance consistently drops when InstantID is used as the test generator**, regardless of the training source.
>
> Upon closer examination, we believe a major reason lies in the unique nature of InstantID. As an identity-preserving generation method, InstantID retains facial identity characteristics from the reference (pristine) image, resulting in a feature distribution that closely resembles that of pristine images. This makes it inherently more difficult to distinguish from authentic images.
>
> Furthermore, we observed that some training generators, such as **Kandinsky**, appear more capable of capturing and generalizing to these identity-preserving characteristics, while others like SD3 and Flux are more prone to confusion when encountering InstantID outputs.
>
> ## Response to Weakness 2: Low Accuracy for Small Regions
>
> To investigate why accuracy remains low for small regions (e.g., ears), we conducted a detailed analysis on the partially manipulated samples in our dataset. As shown in Table, we calculated the average proportion of each manipulated region in both the training and testing sets. The distributions are nearly identical, indicating no significant bias between them. For example, **hair** has the largest average proportion (34.50%), while **eyebrows** are the least represented (0.86%).
>
> | Region | Brow | Ear | Hair | Nose |
> |--------|-----|-------|------|------|
> | Train Set (%) | 0.86 | 1.88 | 34.50 | 2.03 |
> | Test Set (%) | 0.92 | 1.93 | 31.74 | 2.09 |
>
> We further compared a variety of state-of-the-art manipulation localization methods and their performances across different regions, especially focusing on smaller areas like brow. Here are selected results for hair/brow ACC values:
>
> | Type| Method | SD1.5(%)↑ train-hair |  SD1.5(%)↑ brow | FLUX(%)↑ train-hair |  FLUX(%)↑ brow |
> |-------------|----------------------|----------------------|----------------|----------------|---------------------|
> | Naive       | Resnet-50 | 0.95            | 0.66      | 0.99           | 0.36     |
> | Naive       | Xception | 0.98   | 0.63 | 0.99    |  0.17    |
> | Naive       | PSCC-Net | 0.82        |  0.70    | 1.00         | 0.98    |
> | Frequency   | SPSL| 0.98 | 0.77  | 0.98     | 0.35   |
> | Frequency   | F3Net | 0.98            | 0.62  | 0.99           | 0.28    |
> | Frequency   | SRM | 1.00        | 0.99  | 1.00       |  0.86    |
> | Frequency   | NPR | 0.97       | 0.84  | 0.90           | 0.81    |
> | Frequency   | MVSS-Net | 0.89           | 0.88    | 0.96       | 0.95     |
> | Frequency   | MMFusion | 0.99     | 0.94  | 0.99         | 0.12  |
> | VLM-Guided  | UniFD| 0.78    | 0.61   | 0.65           |  0.57   |
>
> From these results, we observe that models integrating **frequency-domain knowledge** (e.g., F3Net, SRM) tend to achieve better generalization to small manipulated regions. This suggests that augmenting MLLM-based explanation models with **frequency-aware reasoning** may help compensate for their current shortcomings in fine-grained localization.
>
> We believe this is a promising direction, and we will continue to explore it in future work.
>
> ## Response to Question 1: Robustness to MLLM Hallucinations and Effectiveness of EkCot
>
> ### Our Core Insight
>
> If we want the MLLM to produce explanations that are both credible and beneficial for model optimization, we must first respect the model’s own judgment bias — i.e., what the MLLM *internally considers* to be the most distinguishing features between real and fake images. This insight directly aligns with our objective (Line 165) of **seek forensic features that the model itself can understand to discriminate the authenticity of an image**, instead of imposing externally defined features.
>
> ### How EkCot Addresses Hallucination
>
> To tackle hallucination and guide the model toward more grounded reasoning, we designed the **Description Generation Module (Section 3.2)** based on the following principles:
>
> - We use the **same MLLM** for both explanation generation and fine-tuning, ensuring consistency in the model's knowledge and representation space.
> - Instead of using a single image (which leads to randomness), we **input a pair of real and fake images**, so the model can compare and find meaningful differences.
> - We **aggregate explanations across multiple image pairs** to extract recurring, low-level forensic attributes — which we define as the model’s **judgment bias**.
>
> This approach avoids handcrafted or subjective features, encourages the model to stay within its own representational bounds, and grounds explanation in behaviorally consistent visual evidence. These qualities distinguish EkCot from prior approaches that rely on static or external labels.
>
> ### Empirical Evidence of Robustness
>
> To evaluate whether the extracted **judgment bias** is effective — especially when facing **subtle forgery traces** like in partially manipulated images — we conducted ablation studies. We isolated the judgment bias from the EkCot pipeline and tested its impact on various subsets.
>
> | Method       | VF  | EKCot (w/o JuB) | EKCot (JuB) | SDXL Partial Synthesis | SDXL Full Synthesis | SD2 Partial Synthesis | SD2 Full Synthesis | Kandinsky2.2 Partial Synthesis | Kandinsky2.2 Full Synthesis | SD3     | Flus   | InstantID |
> |--------------|-----|-------|------|------------------|---------------------|------------------------|--------------------|-------------------------------|----------------------------|---------|--------|-----------|
> | VariantA     |     |       | |69.92                  | 99.92               | 51.07                  | 99.97              | 99.56                         | 99.74                      | 97.66   | 99.12  | 99.69     |
> | VariantB     |     |      |✓| 68.00                  | **99.99**           | 47.64                  | **99.98**          | 99.18                         | **99.99**                  | 99.98   | 99.98  | **99.84** |
> | VariantC     | ✓   | ✓  | | 74.32              | 99.97               | 64.21              | 99.97              | 99.94                     | 99.97                      | **100.00** | **99.98** | 99.41 |
> | **Ours**     | ✓   |      | ✓|**78.62**              | 99.98               | **66.32**              | 99.97              | **99.96**                     | 99.97                      | 99.98 | 99.97 | 99.57 |
>
>
> *Table: Ablation Study of the VLForgery's forgery description generation component in the detection task. 'VF' , 'EKCot' and 'Jub' represent visual input features in forgery description generation,  extrinsic knowledge-guided chain-of-thought, and Judgment Bias, respectively.*
>
> The results (provided in above Table) show that incorporating judgment bias improves detection performance on partially synthesized images, which are known to exhibit more subtle manipulations.
>
> We appreciate the reviewer’s suggestion and will include additional results and analyses to further highlight these findings in the final version.
>
> ## Response to Question 2: Lower Attribution Accuracy in Full Synthesis and Partial Synthesis Methods (Table 6)
>
> ### 1. Architectural Differences: Convolutional Models vs. Vision Foundation Models (VFMS)
>
> In our study, traditional models, which are based on conventional convolutional architectures, tend to show very high attribution accuracy (around 98–99%) for Full Synthesis (FS) samples. However, their performance significantly drops when dealing with Partial Synthesis (PS) samples, with accuracy as low as 6%. In contrast, our method, which is built on vision foundation models (VFMS), exhibits more **balanced performance** across both FS and PS categories.
>
> ### 2. Explaining the Drop in Accuracy
>
> We believe this drop in accuracy can be interpreted with insights from the paper [1], which discusses the asymmetric detection phenomenon in AIGC forensics. Traditional convolutional models are prone to overfitting to specific forgery patterns in the training data, especially in the FS case, where large-scale synthetic artifacts are easier to capture. However, these models struggle with generalization when encountering subtle or unseen manipulations (e.g., PS samples such as PS_SDXL).
>
> - [1] Effort: Efficient Orthogonal Modeling for Generalizable AI-Generated Image
> Detection
>
> In contrast, VFMS-based methods can better incorporate semantic-level information (e.g., scene context, object structure, identity consistency), enabling the model to form a higher-dimensional decision space that fuses both semantic content and forgery cues. This reduces overreliance on any single artifact or forgery pattern and leads to more robust performance across forgery types.
>
> ## Response to Question 3: Robustness Against Unseen Perturbations
>
> Due to the NeurIPS 2025 rebuttal policy that limits the response to 10,000 characters, we have addressed this concern **under our reply to Reviewer fi2j's Weakness 4**, as the two questions are closely related in nature. We sincerely appreciate your understanding.

---

> > ### Comment · Reviewer_9dbu · 2025-08-07
> >
> > Thank you to the authors for the thoughtful response, provides sufficient explanations, and clarifies the concerns I raised. While there is still room for improvement, the paper offers valuable insights for future work on deepfake detection using VLMs. I will maintain my original rating.

---

> ### Author Response · Authors · 2025-08-07
>
> Thank you again for your insightful comments. We would like to kindly follow up on our rebuttal to see if there are any remaining questions or clarifications needed. We are more than happy to discuss further if anything remains unclear.
>
> We truly appreciate your time and consideration.

---

### Official Review · Reviewer_bT45 · 2025-07-03

**Clarity:** 2
**Significance:** 2
**Originality:** 2
**Rating:** 3
**Confidence:** 4

**Summary:**

This paper introduces a single framework for three fake detection problems: detection, localization and attribution. The paper introduces a dataset, called VLF, which contains partially-synthesized images using attribute masks from MM-CelebA-HQ and fully-synthesized images from prompts generated by GPT-4o. The detection model is based on multimodal LLMs, using a prompting template for fake detection, called EkCot, which combines chain-of-thought with extrinsic knowledge in the form of few-shot examples. The fake detection MLLM was finetuned on the VLF dataset with EkCot prompts and tested on a range of datasets and generative models.

**Questions:**

- Please clarify the methodological novelty of EkCot. How is it different from standard Chain-of-Thought prompting and few-shot prompting in LLMs?

- In multiple steps of the proposed method (e.g., Lines 138, 201, 225), the authors used GPT to generate data and prompts. Can the authors provide more motivation and justification? What is the benefit of using GPT-4o here? How does GPT-4o-generated prompts compare to manually-designed prompts and the prompts generated by other LLMs? In addition, how is GPT-4o prompted in each of these steps?

- In Line 201: "For question’s format, we use ChatGPT-4o to generate a range of question formats, as shown in Fig. 2(e), for example: ‘Is this image real or fake? Can you provide the reasoning behind your judgment?’" Can the authors explain the motivation for generating a range of question formats for the same task (fake detection)? Have the authors compared using a range of question formats vs. a single unified question in terms of detection performance?

- Can the authors provide definitions for the undefined technical terms? For example
  - Lines 178-183: Can the authors clearly define "judgement bias" and "judgement bias descriptors" in this paragraph?
  - Line 189: "Initially, we propose a General Rule template designed to integrate the extrinsic information required for generating desired data." Can the authors clarify what the "General Rule template" is?
  - Can the authors define the acronyms in the Tables (e.g., "FS_SDXL", "PS_Kan2.2", etc. in Table 6)?

- Can the authors consider improving the clarity of Figures 2 and 3? The figure can be less clustered and better organized. In figure 3, the font sizes are disproportionate and some text is obscured by annotations.

**Ethical Concerns:**

["NO or VERY MINOR ethics concerns only"]

**Final Justification:**

After discussion, my concerns on the novelty and contributions of the paper were partially addressed. However, significant improvements in writing are still required. Based on these considerations, I increase my rating from 2 to 3.

**Limitations:**

yes

**Quality:**

2

**Strengths And Weaknesses:**

Strengths:
- The proposed method tackles three challenges, i.e. detection, localization and attribution of generative models, in one unified model, which addresses important problems and is relatively novel compared to conventional detection-only approaches.
- The paper conducts an extensive evaluation across multiple generators (SD, Flux and Kandinsky) and forging types.

Weaknesses:
- The technical novelty is limited. The paper primarily combines existing components (e.g. GPT-4o, vision-language models, CLIP, chain-of-thought, prompt engineering, and MLLMs) into a heavily engineered system, rather than proposing a fundamentally new algorithm or model. The reasoning behind many methodology choices are not clear.
- Although many experimental results are presented, insights and new findings are lacking.
- The proposed framework relies heavily on closed-source models (e.g., GPT-4o) and substantial prompt engineering, raising questions on reproduction and generalizability of the ideas.
- The writing quality can be significantly improved. Many technical terms and acronyms are undefined. The clarity of figures can also be improved.

---

> ### Author Rebuttal · Authors · 2025-07-30
>
> I’d like to share some of our thoughts here in hopes of continuing this discussion and eventually reaching a common understanding.
>
> ## Response to Weakness 1 and Question 1: Limited Novelty and Differences from Existing CoT/Few-shot Methods
>
> We humbly disagree with the claim that our work simply combines existing components into a heavily engineered system.Instead, we propose a novel framework that **fine-tunes the MLLM by injecting EkCot knowledge**, enabling it to perform **detection, localization, and attribution** of AI-generated forgeries in a unified manner, rather than relying on auxiliary classifiers or handcrafted prompts.
>
> To compensate for the limitations of existing datasets, we **curated a new dataset from scratch**, focusing on a diverse range of forgery types. Importantly, **GPT-4o was used solely as an auxiliary tool during the dataset creation process**, helping to generate descriptions. It was **not involved at all in our model architecture, inference pipeline, or explanation generation**.
>
> ### As for EkCot, its motivation and mechanism are quite different from previous chain-of-thought or few-shot prompting strategies:
>
> 1. Many previous works [1-5] in explainable AIGC forensics suffer from either **(1)** subjective, ambiguous annotations (often generated by ChatGPT or human annotators), or **(2)** many works use ChatGPT to generate explanatory captions, and then fine-tune another model (e.g., LLaVA or LLaMA) with them. However, these models may have different internal representations and biases, and forcing one model to learn another model’s reasoning maybe harms the forensic performance.
> .
> - [1] DDVQA
> - [2] FFAA
> - [3] x2-DFD
> - [4] FakeShield
> - [5] ForgerySleuth
>
> 2.These methods typically input only a single image when generating explanations. This is problematic because：
>
> - Most open-source MLLMs are primarily tuned for high-level semantic understanding rather than forensic reasoning.
> - AIGC images today are of extremely high visual quality, making superficial or semantic-level explanations uninformative.
> - More importantly, relying on a single image forces the model to hallucinate plausible explanations without a grounded basis, since current MLLMs lack inherent forensic capabilities. In such cases, the model may fabricate reasons for why an image is fake, especially when no real reference is provided.
>
> ### Our Approach
>
> To address this — how to generate trustworthy explanations.
>
> Our key insight: If we want the MLLM to generate explanatory information that is both beneficial for the model's own training optimization and maintains a certain degree of credibility, we must **respect the model’s own judgment bias** — i.e., what it believes are the most distinguishing features between real and fake images.
>
> So, we designed the Description Generation Module (Sec. 3.2) with the following principles:
>
> - We **use the same MLLM** for explanation generation and fine-tuning.
> - Instead of using a single image (which leads to randomness), we **input a pair of real and fake images**, so the model can compare and find meaningful differences.
> - We **aggregate** the outputs across multiple pairs to extract dominant **low-level differences**, which we define as the model’s judgment bias.
>
> This avoids subjective labels, respects model reasoning, and grounds explanations in consistent behavior — distinguishing our method from prior work.
>
> ---
>
> ## Response to Weakness 2: Lack of Insights and New Findings
>
> We would like to clarify that several insights and novel contributions are discussed throughout our other responses:
>
> 1. **A newly constructed explainable AIGC forensic dataset**
>    - Face-focused.
>    - Contains partial and fully synthetic images
>
>    - It contains long prompts and template-based facial prompts, designed to generate diverse and realistic content.
>    - It is annotated with explanation-level labels.
>
> 2. **A unique strategy to extract judgment bias from MLLMs**
>    We propose a Low-Level Feature Comparison Pipeline to mine the model’s internal judgment cues. This aligns the explanation process with what the model itself perceives, rather than relying on external interpretation.
>
> 3. **Exploration of MLLMs' ability to localize manipulated regions through language**
>    We provide qualitative and quantitative analyses showing that MLLMs can not only make binary real/fake decisions but also generate fine-grained textual descriptions pointing to manipulated areas. This opens a new direction for text-based localization in multimodal forensics.
>
> ---
>
> ## Response to Weakness 3 :Reliance on Closed-Source Models and Substantial prompt engineering.
>
> In our framework, GPT-4o is not involved in the model’s training or inference pipeline.
>
> As for the concern regarding prompt engineering: it’s important to understand that **prompt design is a central challenge in explainable multimodal forensics**. Generating reliable and faithful explanations from MLLMs inherently requires thoughtful prompt strategies.
>
> In fact, all recent works [1-6] in this area rely heavily on handcrafted or dataset-specific prompts. For example, [6] introduces an additional classifier to improve detection performance, yet still falls back on traditional prompt-based explanations without fundamental changes. Without meaningful innovation in prompt formulation or explanation mechanisms, the field of MLLM-based forensics cannot truly progress.
>
> - [6] KFD
>
> ---
>
> ## Response to Weakness 4: Comment on Writing Quality, Terminology, and Figures
>
> Thank you for your feedback. We will revise the paper to explicitly define all previously undefined technical terms and abbreviations.
>
> ---
>
> ## Response to Question2: Use of GPT in Multiple Steps (Lines 138, 201, 225)
>
> ### Why Use GPT-4o in Dataset Construction?
>
> In our dataset, some partially manipulated images are generated by modifying specific regions of real images. In such cases, the visual quality and structure of the base image are preserved, and prompting quality has limited impact on visual realism. Using GPT-4o to generate descriptive prompts in this step allows for:
>
> - Increased diversity in the types of image modifications.
> - Semantic richness, since GPT-4o can create more varied and natural attribute combinations.
> - Better coverage of edge cases, making the dataset more comprehensive and robust.
>
> ### Why Use Manually Designed Prompts for Fully Synthetic Images?
>
> In contrast, fully synthetic images (especially human faces) are extremely sensitive to prompt design. Randomly generated prompts from GPT-4o often lead to unstable or low-quality results due to lack of control over critical attributes.
>
> Therefore, we chose to use hand-crafted prompts specifically tailored to generate high-quality synthetic human faces. These prompts were inspired by real-world image generation use cases (popular prompt styles from YouTube creator communities).
> We designed our prompts carefully to ensure high-quality and diverse image outputs, as supported by visual results in our paper.
>
> ### Comparison with Other LLMs and Manual Prompts
>
> While we used GPT-4o in our implementation, we would like to emphasize that other LLMs (e.g., GPT-3.5, Claude, Gemini, etc.) could also be used for generating image-related prompts, as they all have access to rich linguistic and world knowledge. We simply chose GPT-4o based on convenience.
>
> ## Response to Question3:Use of GPT-4o on Line 201, 225
>
> ### Motivation for Using a Range of Question Formats
>
> Inspired by [7] — which showed that different prompt wordings can lead to noticeable performance differences when using LLMs (e.g., ChatGPT) for real vs. fake image detection. Inspired by this, we followed a similar strategy and used ChatGPT-4o to generate multiple semantically consistent but syntactically diverse question formats.
>
> - [7] Can ChatGPT Detect DeepFakes?
>
> Our goal was to improve the overall robustness and generalization of the MLLM's forensic reasoning ability.
>
> However, in our experiments, we found that while using a range of formats does bring slight variation, the impact of question phrasing was relatively minor, especially when compared to other components in the pipeline. Even using a single unified question led to comparable detection performance. Therefore, we decided not to emphasize this part in the main text.
>
> ---
>
> ## Response to Question4: Undefined Technical Terms
>
> ### Judgment Bias and Judgment Bias Description (Lines 178–183)
>
> - Judgment Bias refers to the model’s internal reasoning cu* when determining whether an image is real or fake.
>
>   For example, if we ask ChatGPT: *“This image is fake — can you tell me why?”* and it responds: *“Because it looks blurry,”*
>
>   then the keyword "blurry" is ChatGPT’s judgment bias.
>
> - Judgment Bias Description refers to the generated explanatory statement that incorporates one or more judgment biases in natural language form.
>
>   That is, the judgment bias is only the core evidence (e.g., "blurry"), but the description is the full sentence expressing it in an interpretable way.
>
> We also want to clarify a writing ambiguity in our current draft:
>
> The "specific template" mentioned in Line 183 is the same as the "general rule template" mentioned in Line 189.
>
> ### Acronyms in Table 6 (e.g., "FS\_SDXL", "PS\_Kan2.2")
>
> These acronyms follow the same naming rule as described in the caption of Table 4:
>
> - FS\_SDXL - fully synthetic images generated by the Stable Diffusion XL (SDXL) model.
> - PS\_Kan2.2 - partially manipulated images generated using the Kandinsky 2.2 model.
>
> ## Response to Question 5: Clarity Issues in Figure 2 and Figure 3
>
> Unfortunately, due to NeurIPS rebuttal policy restrictions, we are unable to include revised figures or provide external links at this stage. We appreciate your understanding. We will update these figures in a future version of the paper.

---

> > ### Author Response · Authors · 2025-08-06
> >
> > We sincerely appreciate your time and comments on our paper. We have submitted our rebuttal and would like to kindly check if there are any remaining concerns or points that are unclear. We are happy to provide further clarification or engage in further discussion if needed.
> >
> > Thank you again for your valuable feedback!

---

> > ### Comment · Reviewer_bT45 · 2025-08-07
> >
> > Dear authors,
> >
> > I sincerely appreciate your efforts in addressing my questions. While the rebuttal has addressed some of my questions, I remain concerned about the overall clarity and presentation of the paper.
> >
> > Although the paper may be readable to researchers particularly familiar with MLLM methods for deepfake detection, it is, in my opinion, difficult to follow for readers working on other detection methods or on generative models in general. To improve clarity, I'd suggest replacing the vast amount of implementation details in the methodology section with more explanations on the novelty and design motivations of the proposed method, especially about EkCot and judgement bias filtering.
> >
> >
> > Furthermore, the writing style can be significantly improved to better align with conventions of scientific writing. The paper currently reads more like a technical report with a more casual and conversational tone. Below are some examples:
> >
> > - Line 158: We provide a detailed low-level vision analysis along with all the details and data **you** may need (refer to Sec C).
> >
> > - Line 214: In cases involving multiple edits (e.g., **hairAndnose**), each forged region is localized.
> >
> > - Line 177: These tokens, along with ... are then fed into the **$VLLM()$** for generating ...
> >
> > - Line 193: The ground truth encompasses several categories, including real/fake, forgery type, forgery region, and forgery method. **For pristine images, the forgery type, forgery region and forgery method values are set to ’none’ by default.**
> >
> > - Many undefined acronyms, as pointed out in the original review.
> >
> > - Abuse of underscores. Lines 92, 97, 103, 160, 178, 201, 205, 208.
> >
> > - Inconsistent naming: "ChatGPT", "GPT-4o", "ChatGPT-4o".
> >
> > To meet the standard of a rigorous scientific constribution, I believe substantial improvements in writing are needed.

---

> > > ### Author Response · Authors · 2025-08-07
> > >
> > > We are pleased to hear that the concerns raised in your initial review have been addressed. Your recognition of the improvements is greatly appreciated.
> > >
> > > If there are any remaining issues or points that you feel were not sufficiently clarified in our rebuttal, we would be more than happy to provide further explanation.

---

> > > ### Author Response · Authors · 2025-08-07
> > >
> > > Regarding the new concerns raised in your latest comment, we acknowledge that certain parts of the writing, particularly the clarity and structure of the methodology section, could benefit from further refinement to better serve readers from broader communities beyond MLLM-based detection.
> > >
> > > And we will revise the methodology section to place greater emphasis on the design motivations and novelty of our approach, especially EkCot and judgment bias filtering, and reduce the focus on low-level implementation details. This should significantly improve accessibility and clarity.
> > >
> > > We also understand that some stylistic choices (e.g., use of underscores) may come across as unconventional. Our original intent was to highlight key sections and improve structural clarity, but we agree that this may have impacted the overall academic tone. We will carefully revise these parts and ensure that the writing adheres more closely to scientific writing standards.
> > >
> > > Thank you once again for your helpful input. We will incorporate all these suggestions into the manuscript. If there are any further concerns or questions, please don’t hesitate to contact us.

---

> > > > ### Comment · Reviewer_bT45 · 2025-08-08
> > > >
> > > > Thank the authors for addressing my questions. With remaining concern on the writing quality, I will increase my rating from 2 to 3.

---

> ### Author Response · Authors · 2025-08-07
>
> We hope this message finds you well. We would like to gently follow up on our previous response to your review. If there are any remaining questions or if any part of our rebuttal is unclear, we would be more than happy to provide further clarification.
>
> Thank you again for your time and consideration. We truly appreciate your efforts.

---

### Official Review · Reviewer_ZgwX · 2025-07-14

**Clarity:** 4
**Significance:** 3
**Originality:** 3
**Rating:** 5
**Confidence:** 4

**Summary:**

The paper introduces VLForgery, a new triad framework for AI-generated face forensics that integrates Multimodal Large Language Models (MLLMs) to perform three key tasks: detection, localization, and attribution of facial forgeries.
Unlike traditional binary detectors, VLForgery provides fine-grained analysis by identifying manipulated regions and attributing them to specific diffusion models. Central to this framework, authors introduce the VLF dataset, a large, diverse collection of partially and fully synthesized facial images generated using various diffusion models. The authors also propose EkCot (Extrinsic knowledge-guided Chain-of-thought), a method that enhances MLLMs' reasoning by incorporating low-level visual discrepancies and generation metadata.
Extensive experiments demonstrate that VLForgery outperforms state-of-the-art methods in detection accuracy, localization precision, and attribution reliability, especially in challenging partial synthesis scenarios.

**Questions:**

The paper demonstrates strong performance on the VLF dataset, but how well does VLForgery generalize to real-world or previously unseen forgery methods, especially traditional face-swapping Deepfakes?
You can expand a bit more your future work by providing additional experiments or discussion on cross-domain generalization—e.g., training on VLF and testing on external datasets like Celeb-DF or FaceForensics++. This would help assess the robustness of the model beyond the curated VLF dataset. It could be a demonstration of generalization to real-world or unseen forgery types would significantly strengthenyour work.

Attribution accuracy for partial synthesis is notably lower than for full synthesis (e.g., <70%). What are the main causes of this gap, and how might the model be improved to address it?
Consider providing a breakdown of attribution errors by region or generator, and discuss whether the low performance is due to insufficient signal, dataset imbalance, or model limitations. Suggestions for architectural or training improvements would be valuable.

**Ethical Concerns:**

["Major Concern: Data privacy, copyright, and consent"]

**Limitations:**

Based on the limitations discussed in Section A.1 of the paper,
What strategies can be developed to enhance cross-model generalization, particularly for detecting forgeries generated by previously unseen or novel diffusion models and traditional face-swapping techniques?
How can the VLF dataset be expanded or diversified to better capture the semantic variability introduced by different generators and prompts, and how does this variability affect detection and attribution performance?
It could be nice to read some lines concerning these points.

**Paper Formatting Concerns:**

Congratulations, I think it is a good quality paper, including formatting.

**Quality:**

4

**Strengths And Weaknesses:**

Here is a list of identified strengths:
- The proposed integration of MLLMs with structured reasoning (EkCot).
- The VLF dataset is comprehensive, covering multiple synthesis types and generators.
- The framework achieves state-of-the-art performance across multiple forensic tasks.
- The use of natural language for localization enhances interpretability and usability.
- The paper includes thorough ablation studies and qualitative comparisons.
some weaknesses are:
- Generalization to unseen generators and real-world forgeries remains an open opportunity, especially for partial synthesis.
- The reliance on handcrafted low-level vision biases and manual filtering introduces subjectivity.
- The localization accuracy for small regions (e.g., ears) is still relatively low.
- The framework’s performance on traditional Deepfakes is not evaluated; it could limit its broader applicability.
- The computational cost of training and inference with large MLLMs is not deeply discussed, it is an open area.

---

> ### Author Rebuttal · Authors · 2025-07-27
>
> Thank you for your recognition of our work and the paper’s formatting quality. We appreciate your positive feedback.
>
> ## Response to Weakness 1: Generalization to Unseen Generators and Real-World Forgeries
>
> We appreciate the reviewer’s observation. Indeed, our motivation in constructing the VLF dataset was driven by the limitations we observed in existing face forgery datasets at the time, particularly with respect to partial synthesis using diffusion models. Most available datasets had limited partial forgeries and were often of **low visual quality**, largely due to the lack of carefully crafted prompts tailored to face synthesis.
>
> For instance, we examined the dataset from [1] and found that many generated faces exhibited significant distortion and artifacts. In response, we developed a set of long, diverse, and high-quality prompt templates—partially inspired by observations from a YouTube channel (unfortunately, we are unable to disclose this due to policy restrictions). As shown in Section B.4, our generated face images achieve notably higher visual fidelity.
>
> - [1] Diffusion Facial Forgery Detection
>
> We acknowledge that generalization to unseen generators and real-world forgeriesis an important open direction. Given the emergence of newer datasets and generators in recent months, we plan to conduct broader generalization studies in future work. However, due to limited computational resources and time constraints, a full-scale evaluation on all unseen generators and real-world samples could not be completed for this submission.
>
>
> ## Response to Weakness 2: Subjectivity in Handcrafted Low-Level Vision Biases
>
> We acknowledge that this approach inevitably introduces a degree of subjectivity, especially in the filtering and bias selection process.
>
> Inspired by prior work [2], which explores specialized models for low-level vision comparison (e.g., image quality assessment or structural similarity), we believe a promising future direction would be to incorporate multi-expert models or objective evaluation networks to generate more reliable and less subjective low-level vision biases.
>
> - [2] Towards Open-ended Visual Quality Comparison
>
> We recognize that this would be a significantly more complex engineering effort, but we see it as a valuable path forward, and we are committed to further exploring this direction in future work.
>
> ## Response to Weakness 3: Low Localization Accuracy for Small Regions
>
> To investigate why localization accuracy remains low for small regions (e.g., ears), we conducted a detailed analysis on the partially manipulated samples in our dataset. As shown in Figure, we calculated the average proportion of each manipulated region in both the training and testing sets. The distributions are nearly identical, indicating no significant bias between them. For example, **hair** has the largest average proportion (34.50%), while **eyebrows** are the least represented (0.86%).
>
> | Region | Brow | Ear | Hair | Nose |
> |--------|-----|-------|------|------|
> | Train Set (%) | 0.86 | 1.88 | 34.50 | 2.03 |
> | Test Set (%) | 0.92 | 1.93 | 31.74 | 2.09 |
>
> We further compared a variety of state-of-the-art manipulation localization methods and their performances across different regions, especially focusing on smaller areas like brow. Here are selected results for hair/brow ACC values:
>
> | Type| Method | SD1.5(%)↑ train-hair |  SD1.5(%)↑ brow | FLUX(%)↑ train-hair |  FLUX(%)↑ brow |
> |-------------|----------------------|----------------------|----------------|----------------|---------------------|
> | Naive       | Resnet-50 | 0.95            | 0.66      | 0.99           | 0.36     |
> | Naive       | Xception | 0.98   | 0.63 | 0.99    |  0.17    |
> | Naive       | PSCC-Net | 0.82        |  0.70    | 1.00         | 0.98    |
> | Frequency   | SPSL| 0.98 | 0.77  | 0.98     | 0.35   |
> | Frequency   | F3Net | 0.98            | 0.62  | 0.99           | 0.28    |
> | Frequency   | SRM | 1.00        | 0.99  | 1.00       |  0.86    |
> | Frequency   | NPR | 0.97       | 0.84  | 0.90           | 0.81    |
> | Frequency   | MVSS-Net | 0.89           | 0.88    | 0.96       | 0.95     |
> | Frequency   | MMFusion | 0.99     | 0.94  | 0.99         | 0.12  |
> | VLM-Guided  | UniFD| 0.78    | 0.61   | 0.65           |  0.57   |
>
> From these results, we observe that models integrating **frequency-domain knowledge** (e.g., F3Net, SRM) tend to achieve better generalization to small manipulated regions. This suggests that augmenting MLLM-based explanation models with **frequency-aware reasoning** may help compensate for their current shortcomings in fine-grained localization.
>
> We believe this is a promising direction, and we will continue to explore it in future work.
>
> ## Response to Weakness 4 and Question 1: Lack of Evaluation on Traditional Deepfakes
>
> We acknowledge that the current version of our framework has not extensively evaluated performance on traditional Deepfake datasets, which may limit the generalizability of our conclusions.
>
> Due to limited time and computational resources, we were unable to construct a separate training and testing pipeline on large-scale Deepfake datasets. However, as a preliminary effort, we evaluated our model on the face-swapping subset of FaceForensics++ .
>
> | Train | DF    | F2F   | FS    | FST   | NT    | Avg(%)↑ |
> |-------|-------|-------|-------|-------|-------|---------|
> | DF    | 96.11 | 96.34 | 95.74 | 95.98 | 95.90 | 96.01   |
> | F2F   | 98.79 | 99.09 | 98.63 | 98.38 | 98.47 | 98.67   |
> | FS    | 99.05 | 99.20 | 99.05 | 98.99 | 99.05 | 99.07   |
> | FST   | 97.09 | 97.30 | 96.57 | 97.01 | 97.01 | 96.99   |
> | NT    | 99.72 | 99.70 | 99.56 | 99.60 | 99.62 | 99.56 |
>
> *Table: Evaluation on Traditional Face-swapping Deepfakes type. Results on different training and testing subsets using VLForgery. Accuracy is used for evaluation.*
>
> We agree that a more comprehensive study involving diverse Deepfake sources (e.g., face reenactment, lip-syncing, expression manipulation) would be valuable.
>
>
> ## Response to Weakness 5: Computational Cost of Large MLLMs
>
> Thank you for pointing this out — we acknowledge that we did not provide sufficient details regarding the computational cost.
>
> In our current setup, training the model for one epoch using 8× NVIDIA RTX L40 GPUs takes approximately 6–8 hours. During inference, generating 100 tokens for a single image takes about 3–5 seconds on a single RTX L40.
>
> ## Response to Question 2: Attribution Accuracy Gap Between Partial and Full Synthesis
>
> We believe this issue is closely related to *Weakness 3*, which concerns low localization accuracy for small modified regions.
>
> Our hypothesis is that attribution fundamentally relies on detecting *consistent semantic patterns* across an image. In the case of **full synthesis**, all regions of the image are generated by the same model and thus share a **unified generative signature or semantic paradigm**. However, for partial synthesis, only a small localized region follows the generative model's semantic pattern, while the rest remains real. This **heterogeneity** makes the attribution task more challenging.
>
> As mentioned in our response to Weakness 3, incorporating additional frequency-aware features may help enhance the model’s ability to detect and attribute subtle manipulations. These features have shown promise in improving generalization to small forgery regions.
>
> We also acknowledge the reviewer's point regarding possible dataset imbalance and insufficient signal. While we have balanced the number of examples per generator type, there is an intrinsic imbalance in region size and semantic coverage between full and partial synthesis samples, which could degrade attribution performance.
>
> Since few prior works directly study **attribution for partial synthesis**, our current findings represent a preliminary step.
>
> We appreciate the reviewer’s valuable suggestions and will consider these directions in future work.

---

> ### Comment · Reviewer_ZgwX · 2025-08-05
>
> I appreciate the authors' thoughtful rebuttal and the additional experiments provided. These responses satisfactorily address the concerns I raised in my initial review. After considering the reviews and the authors' clarifications, my decision is still Accept.
> Thank you for your efforts.

---

> > ### Author Response · Authors · 2025-08-06
> >
> > We sincerely appreciate your time and effort in reviewing our rebuttal and providing final feedback. Your insights have been valuable to us, and we are grateful for your constructive comments throughout the review process.

---

### Decision · Program_Chairs · 2025-09-17

**Decision:**

Accept (poster)

**Comment:**

The paper introduces VLForgery, a triad framework for AI-generated face forensics that integrates MLLMs to perform three key tasks: detection, localization, and attribution of facial forgeries. Compared to traditional binary detectors, VLForgery provides fine-grained analysis by identifying manipulated regions and attributing it to specific diffusion models. The authors also present the VLF dataset by employing GPT4o and various diffusion models to generate forged face images. Additionally, the paper introduces the EkCot (Extrinsic knowledge-guided Chain-of-thought) method, which combines external knowledge from the image generation pipeline with low-level visual differences to guide MLLMs in performing credible forgery analysis.

Overall speaking, the proposed work is well-motivated, with moderate novelty, technically sound, with extensive experiments, and reporting superior results. More importantly, the work advances the field from mostly binary classification to detailed analysis with some degree of explainability. The new dataset is also good for the field to shift to new tasks.

There are some negative concerns mainly on generalizability to real-world unseen forgery methods, relatively low accuracy on forgery localization, heavy reliance on closed-source models (GPT-4o), lack of evaluations on some perturbations, and writing quality issue. During the rebuttal, most of the concerns have been solved by the authors. One reviewer is not satisfied mainly due to the writing problem. While indeed there are many writing problems, they do not downgrade the contribution of the work essentially. Therefore, the AC suggests an Accept. In the final version, the authors should improve the writing very carefully by considering the reviewer’s suggestions.